# Orbit related sea level errors for TOPEX altimetry at seasonal to decadal time scales

Saskia Esselborn[1], Sergei Rudenko[1,2], Tilo Schöne[1]

[1]GFZ German Research Centre for Geosciences, Department 1: Geodesy, Potsdam, 14473, Germany
[2]Deutsches Geodätisches Forschungsinstitut (DGFI-TUM), Technische Universität München, Munich, 80333, Germany (since August 2016, until that – at GFZ)

*Correspondence to*: Saskia Esselborn (Saskia.Esselborn@gfz-potsdam.de)

**Abstract.** Interannual to decadal sea level trends are indicators of climate variability and change. A major source of global and regional sea level data is satellite radar altimetry, which relies on precise knowledge of the satellite's orbit. Here, we assess the error budget of the radial orbit component for the TOPEX/Poseidon mission for the period 1993 to 2004 from a set of different orbit solutions. The errors for seasonal, interannual (5 years), and decadal periods are estimated on global and regional scales based on radial orbit differences from three state-of-the-art orbit solutions provided by different research teams (GFZ, GSFC, and GRGS). The global mean sea level error related to orbit uncertainties is of the order of 1 mm (more than 10 % of the global mean sea level variability) with negligible contributions on the annual and decadal time scales. In contrast, the orbit related error of the interannual trend is 0.1 mm/year (18 % of the corresponding sea level variability) and might hamper the estimation of an acceleration of the global mean sea level rise. For regional scales, the gridded orbit related error is up to 11 mm and for about half the ocean the orbit error accounts for at least 10 % of the observed sea level variability. The seasonal orbit error amounts to 10 % of the observed seasonal sea level signal in the Southern Ocean. At interannual and decadal time scales, the orbit related trend uncertainties reach regionally more than 1 mm/year. The interannual trend errors account for 10 % of the observed sea level signal in the Tropical Atlantic and the south-eastern Pacific. For decadal scales, the orbit related trend errors are prominent in a couple of regions including: South Atlantic, western North Atlantic, central Pacific, South Australian Basin, and Mediterranean Sea. Based on a set of test orbits calculated at GFZ, the sources of the observed orbit related errors are further investigated. Main contributors on all time scales are uncertainties in Earth's time variable gravity field models and on annual to interannual time scales discrepancies of the tracking station sub-networks, i.e., SLR and DORIS.

 **1 Introduction**

Sea level is an important indicator of climate variability and change. Based on tide gauge data using different techniques, the global mean sea level rise for the last century is estimated to be 1.2-1.9 mm/year (Douglas, 1997; Church and White, 2011; Jevrejeva et al., 2008, 2014; Hay et al., 2015). Based on satellite altimetry data since 1993, the current rate of global mean sea level has been estimated to be more than 3 mm/year (Cazenave et al., 2014; Ablain et al., 2016, Quartly et al., 2017). The main sources of the current rise are thermal expansion of the sea water and melting of glaciers and ice sheets. At interannual time scales, changes of terrestrial water storage imprint additionally on the global mean sea level (Llovell et al., 2011). Recent work (Watson et al., 2015; Fasullo et al., 2016) has focussed on the detectability of accelerations in global mean sea level trends during the last decades. Regionally, sea level rates during the last 24 years show higher variability, they range from -1 mm/year to more than 10 mm/year. They are mainly linked to regional changes in the oceans density field, which might be induced by internal ocean variability, atmosphere-ocean interaction, or influx of freshwater. Satellite altimeters are a unique source of global and regional sea level data and are available continuously since the beginning of the 1990s. Precise orbits of altimetry satellites are a precondition for global and regional mean sea level investigations (Rudenko et al., 2012; Rudenko et al., 2014) and errors related to precise orbit determination (POD) are demonstrably one of the major error sources for global and regional sea level products (Ablain et al., 2015). A detailed description of the main factors contributing to the radial orbit errors is given by Fu and Haines (2013). The orbit errors have typically long wavelengths and may contain systematic contributions at seasonal to decadal timescales.

Couhert et al. (2015) investigated the main contributions to the radial orbit error budget for the Jason-1 and Jason-2 series based on Geophysical Data Records (GDR)-D at seasonal to decadal time scales for the second altimetry decade (2002-2013). According to their analysis, the orbit related uncertainty of the global mean interannual and decadal trends is less than 0.1 mm/year. As main factors for regional errors they identified contributions from tracking data and from reference frame (up to 8 mm) at seasonal time scales, contributions from tracking data (up to 3 mm/year) and Earth's time variable gravity field (up to 2 mm/year) at interannual time scales, and contributions from tracking data (up to 2 mm/year) and Earth's time variable gravity field (up to 1.5 mm/year) at decadal time scales. A correspondent assessment for the first altimetry decade (1992-2001) has still been missing and is the rationale of this paper.

We assess the error budget of the radial orbit component for the TOPEX/Poseidon mission for the period 1993 to 2004 from a set of different orbit models. We have chosen TOPEX/Poseidon, since it is the reference altimetry mission used in the European Space Agency's (ESA) Climate Change Initiative (CCI) Sea Level project over this time span (Ablain et al., 2016). We assess the radial orbit error budget at regional and global scales at seasonal, interannual, and decadal time scales by the analysis of three state-of-the-art orbit solutions derived and provided by different research teams from the German Research Centre for Geosciences (GFZ), the Groupe de Recherche de Geodesie Spatiale (GRGS), and the Goddard Space Flight Centre (GSFC). Note, that our assessment necessarily excludes contributions from errors common to these three orbits. However, since the three orbits were derived using various up-to-date models, the errors common to the three orbits should be rather low which makes us confident that our error estimates represent most of the error. In our further analyses,

we use test orbits calculated at GFZ to investigate the impact of uncertainties of the tracking station sub-networks, of the reference frame, and of the Earth's time variable gravity field models on the radial orbit component, and hence the derived sea level.

A detailed description and assessment of the analysed orbits as well as specifications of the altimeter data processing are given in Sect. 2. Sect. 3.1 describes the methods implemented to assess the orbit errors for the different time scales and the corresponding results for global and regional scales. The estimates of the orbit related error for global mean and regional sea level are given in Sect. 3.2 and 3.3, respectively. The specific orbit related errors for ascending and descending passes are investigated in Sect 3.4. In Sect 3.5 we examine for which areas the orbit error reachs more than 10 % of the corresponding sea level varalibility. The main findings are summarized and discussed in Sect. 4.

## 2 Orbit and altimetry data

### 2.1 Description of the analysed orbit solutions

Our aim is to assess the range and the characteristics of radial orbit errors on regional and global scales. Therefore, the differences between three independent state-of-the-art orbit solutions available for the TOPEX/Poseidon mission are analysed. All orbit solutions are derived in the International Terrestrial Reference Frame (ITRF) 2008 reference frame (Altamimi et al., 2011) and use Satellite Laser Ranging (SLR) and Doppler Orbitography and Radiopositioning Integrated by Satellite (DORIS) tracking data, but are based on different software and on distinct models. The actual multi-mission GFZ orbit solution VER11 (Rudenko et al., 2017) is used as a reference in this paper and is called REF hereafter. The GSFC std1504 orbit (Lemoine et al., 2010; Beckley et al., 2015) has been chosen by the ESA CCI Sea Level Phase 2 project and differs in many aspects from the GFZ orbit, regarding software as well as the suite of implemented models including another Earth's gravity field model. As the third model, we have chosen the GRGS orbit solution (Soudarin et al., 2016), which is derived using models similar to those of the GFZ solution, but employing another software package. The main models used for GFZ REF, GRGS, and GSFC std1504 orbits are described in Table 1. The main differences in these three orbit solutions are related to the choice of the Earth's time variable gravity (TVG) field models, ocean tide model, modelling of non-tidal atmospheric and oceanic gravity, the treatment of geocenter variations in station displacements as well as the constraints of the observation data (SLR/DORIS). While for the GRGS solution comparatively high weight is on the SLR data, for the GFZ solution there is higher weight on the DORIS data. Proper modelling of the Earth's gravity field, in particular of its time-variable part, is crucial for the computation of precise orbits of altimetry satellites and has been shown to contribute to errors in regional sea level trends and seasonal signals (Rudenko et al., 2014; Esselborn et al., 2015). For the pre-GRACE period the TVG field is poorly constrained. The weekly TVG solutions used for the GSFC orbit were derived up to degree and order 5 from the analysis of SLR and DORIS observations to 20 geodetic satellites starting from 1993 (Lemoine et al., 2016). The TVG part used for the GFZ REF (GRGS) orbits consists of the combination of yearly coefficients, drift terms and annual and semi-annual variations for degree and order 1 to 80 (2 to 50) derived from GRACE data and SLR measurements

to LAGEOS-1/2. The annual and semi-annual coefficients used for the GFZ REF orbit are fitted yearly starting from August 2002. For the pre-GRACE period before August 2002 (January 2003) only the degree 2 terms exhibit yearly values and drift terms, however, the annual and semi-annual variations, which were derived for the GRACE-period, are applied for degree and order 1-80 (2-50) (Rudenko et al., 2014, Förste et al., 2016).

The approach adopted for the estimation of the radial orbit errors implicates that errors common to all three orbits can not be detected. In particular, all three orbits rely on the ITRF2008 reference frame and basically the same set of tracking stations. To further estimate the orbit related radial orbit error budget due to the most significant factors, we have derived five test orbits based on the GFZ REF orbit. The errors related to inconsistencies of the tracking data networks are tested by using only one tracking network instead of two. Since the GRGS orbit was derived without estimation of the DORIS system time

bias, we have studied the impact of this bias on the radial orbit differences with special focus on systematic differences between ascending and descending passes. The effect of errors in the realization of the terrestrial reference frame is tested by the implementation of the most recent ITRF2014 version. The effects of uncertainties in Earth's TVG field models are tested by the implementation of the EIGEN-6S2 model which is the predecessor of the EIGEN-6S4 model. For each case, the same background models and estimated parameters were used as for the REF orbit, except for those that represent the changes for

the specific test case. The five test orbits and the differences with respect to the GFZ REF orbit are:

- SLR orbit: derived by using SLR tracking observations only,
- DORIS orbit: computed by using DORIS tracking observations only,
- TBias orbit: calculated without estimation of the DORIS system time bias,
- ITRF14 orbit: calculated by using the information on station positions and velocities from ITRF2014 (Altamimi et
al., 2016) instead of ITRF2008,
- Geoid orbit: obtained by using EIGEN-6S2 (Rudenko et al., 2014) Earth's gravity field model instead of EIGEN-6S4 model (Förste et al., 2016). Note that the Geoid orbit is based on the same gravity field model as the GRGS orbit.

**2.2 TOPEX altimeter data**

In order to assess the orbit accuracy at crossover points and to relate the estimated errors to the total variability of the sea level data, along-track TOPEX Sea Level v1.1 ECV data (Ablain et al., 2015) released from the ESA CCI Sea Level project has been included in the analyses. The along-track data has been corrected for all instrumental and geophysical effects by the state-of-the-art models provided with the data. However, for some corrections updated models were applied. These include: EOT11a ocean tides and loading tides (Savcenko and Bosch, 2012), solid earth tides following the IERS 2003 conventions,

and updated GPD+ wet tropospheric corrections (Fernandes and Lazaro, 2016). The altimeter crossover differences were calculated for each test orbit separately. For the calculation of sea level anomaly grids the GSFC std1504 orbits have been selected. The processing of the data, the crossover point and collinear analyses as well as the interpolation to a regular grid were performed using GFZ's Altimeter Database and Processing System (ADS) Central (Schöne et al., 2010).

## 2.3 Evaluation of the orbit solutions

In the following, the performance of the analysed orbits is evaluated. For the GFZ orbit solutions, the consistency with tracking data and at arc overlaps is assessed. Table 2 provides the main results of precise orbit determination of the GFZ reference and test orbits, namely, the average values of SLR and DORIS RMS fits, radial, cross-track, and along-track two-day arc overlaps, illustrating the internal orbit consistency in these directions, and the number of the arcs used to compute these values for the reference and five test orbits. When using the same observation types and weighting, smaller values of

arc overlaps and observation fits indicate improved orbit quality. Reduced radial arc overlaps characterise reduced radial orbit error. SLR observations were used at all 494 orbital arcs of five GFZ orbits, except for the DORIS orbit for which no SLR observations were used at all. Since DORIS data are available for TOPEX/Poseidon only until October 31, 2004, these data were used at 459 orbital arcs preceding this date, except for the SLR orbit for which no DORIS observations were used at all. All orbital arcs for GFZ orbits are manoeuvre-free. Thus, two-day arc overlaps were computed for 433 overlaps for the

REF, TBias, ITRF14, and Geoid orbits. In case of the SLR and DORIS orbits, a few gaps in the observations caused radial arc overlap larger than 0.5 m. Those arc overlaps have been excluded from the statistics resulting in less arc overlaps shown for these orbits in Table 2.

Fig. 1 provides information on the SLR RMS fit of the reference and tests orbits, while Fig. 2 displays the radial arc overlap of two consecutive two-day orbit arcs. The four orbits derived using SLR and DORIS observations provide comparable

levels of average SLR RMS fits (1.96 – 1.99 cm, Fig. 1). The smallest SLR RMS fit (1.59 cm) but largest radial arc overlap (1.72 cm) is obtained for the SLR-only orbit indicating a weak orbit quality over large geographical areas. The largest SLR RMS fit is obtained for the TBias orbit. When no DORIS system time bias is estimated inconsistencies between the timing of the observation system result in higher misfits. Among the five orbits derived using DORIS observations, a slightly increased average value of DORIS RMS fits (0.04795 cm/s) is obtained for the DORIS orbit derived using only DORIS

observations (related to the weighting of observation types and the number of observations used) followed by the TBias orbit (0.04785 cm/s), while the other orbits derived using SLR and DORIS observations (REF, ITRF14, and Geoid) show comparable average values of DORIS RMS fits (0.04775 – 0.04778 cm/s). The smallest average value of the radial arc overlaps (Fig. 2) is obtained using the EIGEN-6S2 geopotential model (0.83 cm). The radial arc overlaps of the TOPEX/Poseidon orbit derived using only SLR data are 1.95 times larger than those of the orbit derived using only DORIS

data. Using the reference frame ITRF2014 instead of ITRF2008 eliminates many outliers in the radial arc overlaps and therefore reduces the average value of the radial overlaps from 0.90 to 0.84 cm.

The DORIS system time bias is regularly estimated and applied during GFZ's POD process to adjust the DORIS time system to the SLR time system. Zelensky et al. (2006) showed that there is a strong linear relationship between along-track orbit position and the DORIS time bias. The comparison of the fits and overlap values of the REF and the TBias orbit (Table 2)

shows that the estimation of the DORIS time bias improves the orbit quality. The temporal behaviour of the DORIS system time bias derived for TOPEX/Poseidon REF, ITRF14 and Geoid test orbits is in close agreement (Fig. S1) and resembles the estimation given by Lemoine et al. (2016). For the GFZ VER11 (REF) orbit, it indicates variations between -22.4

microsecond (μs) and +4.4 μs from 1992.73 to 1994.18, followed by a period of a linear trend of 35.11 μs/year between 1994.18 and 1995.00 that ends with a jump from -28.65 μs to +1.98 μs around 1995.00. Then the DORIS time bias shows two rather stable periods with a mean value of +3.70 μs with a standard deviation of 1.77 μs from 1995.0 to 1999.0 and a mean values -1.32 μs with a standard deviation 1.19 μs from 1999.0 to 2001.13, followed again by a period of a linear trend (-3.14 μs/year) from 2001.13 to 2004.83. The mean value of the DORIS system time bias is 0.04 ± 0.36 μs for the DORIS test orbit, and it is equal to zero (not shown in the figure) for the TBias orbit.

For all orbit solutions, a crossover point analysis for the period April 1993 to September 2004 has been performed based on the altimeter data described in Sect. 2.2. Differences between the values of ascending and descending passes at crossover points are caused by oceanic variability and errors related to the measurements, the orbit, and the applied corrections. Since in our study errors related to the measurements and the applied corrections and oceanic variability are always identical, here, smaller absolute median differences and decreased RMS values at crossover points are indicative for increased orbit quality. The median of the time series of global mean height differences and RMS values at the crossover points are provided in Table 3. The smallest ascending/descending differences (-1.6 mm) and as well the lowest RMS values (49.5 mm) at the crossover points are reached by the GSFC orbit solution. The median global ascending/descending differences are -3.1 mm for the GFZ REF and -3.0 mm for the GRGS orbit solutions. However, while the RMS value of the GFZ REF solution (49.8 mm) is comparable to the one of the GSFC (49.5 mm), the GRGS orbit solution shows degraded performance (51.3 mm RMS). Keeping the DORIS time bias fixed to zero deteriorates the median differences between ascending and descending passes to -3.6 mm, but does not change the RMS value. The median of the global mean ascending/descending differences is -2.7 mm for the SLR and -4.7 mm for the DORIS orbits. Both orbit solutions show degraded performance (51.2/50.7 mm RMS) with respect to the REF solution. This shows that using SLR and DORIS observations together improves the orbit quality considerably, even though the DORIS observations seem to aggravate the mean differences between ascending and descending tracks. Using ITRF2014 instead of ITRF2008 does not change the RMS of crossover differences, but improves their median values. The Geoid orbit solution exhibits clearly improved ascending/descending differences (-2.1 mm) as well as a slight reduction of the RMS values. A further analysis of the temporal evolution of the ascending/descending differences reveals that these improvements take place in the pre-GRACE period before August 2002.

## 3 Estimation of the orbit related sea level error

Sea level is varying on typical temporal and spatial scales, that are often connected to the driving processes. At the same time, orbit errors are not randomly distributed but exhibit also typical temporal and spatial pattern. Here, we apply statistical methods in order to assess the errors related to the orbit solutions for global and regional sea level at seasonal to decadal time scales.

## 3.1 Methods

In order to estimate the orbit related errors in sea level height, the differences between the radial components of the GFZ
REF orbit and the two independent orbit solutions (GSFC and GRGS) have been analysed. To assess the effect of
uncertainties in the reference system, in the realisation of the tracking station networks, and in Earth's time variable gravity
on the radial error budget, we have evaluated the differences of the radial orbit components between the GFZ's REF and
ITRF14, SLR, DORIS, and Geoid test orbits. Since the radial orbit components map directly to the derived sea level heights,
we consider the differences presented here to represent estimates of the orbit related sea level error. However, since the orbit
error analysis is based on orbit differences, any error common to all three orbits will be lacking in our assessment.

The differences of the radial orbit components at the time of the altimetry measurement (1 Hz, ~6.7 km on ground) are
calculated and interpolated to a global 1°x1° grid for every cycle (9.92 days). In general, we merge both, ascending and
descending, passes in our calculations. In addition, we analyse ascending and descending passes for some orbit combinations
separately. In order to study the global mean differences between the radial orbit components and their temporal evolution,
global mean RMS values per cycle are derived. They are calculated as the square root of the spatial weighted mean of all
squared radial orbit differences on the 1°x1° grid for the respective cycle.

Since we are not interested in the orbit error itself, but rather in the effect of radial orbit errors on global and regional sea
level, we treat the radial orbit differences the same way as the sea level values from altimetry. For the estimation of global
mean errors, the gridded radial orbit differences are averaged (with area weighting) over the ocean (±67° latitude). Starting
from these global mean orbit differences, global mean RMS values relative to the temporal mean of the series are calculated
as an estimate for the orbit related error of the global mean sea level. Decadal trends, annual and semi-annual signals, and the
corresponding formal errors are estimated by a least-square fit. The seasonal errors are derived from the amplitudes of the
annual signal. As a measure for errors at interannual time scales, we calculate the RMS of the five-year running trend series
of the radial orbit differences. Since the times series is only 11 years long it is not possible to derives statistically sound
estimates of the decadal trend. Here, the errors of decadal trends are assumed to correspond to the absolute values of the
trends fitted to the series of the radial orbit differences. For the estimation of regional upper bound errors, at each grid point
RMS values relative to the local temporal mean, annual cycle, RMS of the 5-year running trend, and decadal trends are
calculated in correspondence to the global analyses. From the 1°x1° grid the maximum values over the ocean are extracted to
estimate regional upper bound errors.

In order to relate the estimated errors to the total variability of the sea level data, TOPEX altimeter data  has been included as
well. The data and the processing are described in Sect. 2.2. From the gridded sea level anomalies, seasonal, interannual and
decadal trends were derived using the methods described above.

## 3.2. Global mean errors

In the following, we investigate the orbit related global sea level error, differentiating between the total error and its annual, interannual, and decadal components. The TBias orbit differences are not included in these analyses but will be further investigated for the study of changes between ascending and descending passes (Sect. 3.4). The time series of the global mean RMS of gridded radial orbit differences per cycle are shown in Fig. 3 for all orbit solutions relative to GFZ's REF orbit. The largest differences occur between the REF and the GRGS orbits, the smallest changes occur for the ITRF14 test orbit. Most orbit differences are dominated by sub-seasonal variability, only for the Geoid and ITRF14 orbit the RMS per cycle series are governed by seasonal and decadal periods. For the Geoid, GSFC, and GRGS orbit differences relative to the REF orbit, the RMS series exhibit a seasonal cycle, which is an indication for seasonal orbit differences on regional scales. The RMS of the *REF minus Geoid* orbit difference is decreased after August 2002 indicating that the main differences between the two orbits originate from the pre-GRACE period. In contrast, the differences between the REF and the ITRF14 orbits are slightly increasing from 2000 onwards.

From the time series of global mean orbit differences over the oceans, RMS, annual cycle, 5-year trend variability, and decadal trend differences are calculated and used as an estimate of the orbit related error on different time scales. These orbit errors are summarized in Table 4 for all orbit models together with the corresponding values derived from altimetric sea level anomalies. The global mean RMS of the radial orbit differences between the REF and GRGS (GSFC) orbits amount to 1.2 (1.1) mm, which corresponds to more than 10 % of the global mean sea level variability of 10.2 mm. The restriction to one tracking station sub-network leads to large changes of the orbit, for the DORIS (SLR) orbit solution the RMS values of the radial differences with respect to the REF orbit amount to 1.8 mm (0.7 mm), which exeedsthe size of the estimated total orbit errors. This highlights the importance of manifold, precise, and consistent tracking data for accurate global mean sea level estimates. The substitution of the Earth's gravity field model (EIGEN-6S4 by EIGEN-6S2) and the ITRF realization (ITRF2008 by ITRF2014) accounts for 0.2 mm and 0.3 mm, respectively, of the mean RMS orbit errors. A spectral analysis of the global mean radial differences (Fig. S2) exhibits peaks at ~60 days for all but the GRGS and TBias orbit differences and at ~90 and ~170 days for the SLR and DORIS orbit differences. An annual component can be observed for the GRGS and Geoid orbit differences. Since the annual amplitude is less than 1 mm only, it can be neglected and is not included in Table 4. The time series of the five-year running trends of the global mean radial orbit differences over the ocean are shown in Fig. 4 for the various orbit combinations. All curves range between -0.3 mm/year and +0.2 mm/year and show at least one zero-crossing and imply interannual changes of the estimated decadal sea level trends. The corresponding curve of the five-year running trends for the global mean sea level (not shown) range between 4 mm/year at the beginning of the time series and 2 mm/year at the end. Before 1998 the GSFC and GFZ solutions are close to each other and both suggest smaller sea level trends for this period than the GRGS solution. After that, trends derived from GFZ orbits are weaker than the ones derived from GSFC orbits and stronger than the ones derived from GRGS. The maximum interannual trend variability of 0.1 mm/year occurs between the REF and GRGS orbits (Table 4) which amounts to almost 20 % of the corresponding value derived for the global mean sea level curve (0.55 mm/year). An error of this size might interfere with the estimation of

global mean sea level acceleration. Hence, relative to the GFZ orbits the use of the GSFC (GRGS) orbits would result in a slightly increased (decreased) acceleration of the global mean sea level curve during the TOPEX period. Since the exclusive use of DORIS tracking station leads to interannual trend variability of 0.11 mm/year, inconsistencies of the tracking stations sub-networks might explain large portions of the observed global mean interannual variability. The errors of the interannual trend variability are for all orbit combinations higher than for the decadal trends. The global mean decadal trends (calculated over the full mission time) are mostly significant but can be further neglected, since they are well below the uncertainty of the corresponding global mean decadal sea level trend (±0.5 mm/year).

### 3.3 Regional errors

The maximum regional errors derived from the analysis of the gridded orbit difference series over the oceans are summarized in Table 5. The TBias orbit differences are not included in these analyses but will be further investigated for the study of changes between ascending and descending passes (Sect. 3.4). Regionally, the maximum radial orbit differences on the 1°x1° grid between the REF and GRGS (GSFC) orbits amount to 10.7 (7.4) mm. The exclusive use of only one tracking station sub-network leads to distinct changes with RMS values of 9.3 (7.2) mm for the DORIS (SLR) sub-network. This suggests that for the weighting factors applied with GFZ's REF orbit especially inhomogeneity in the SLR station sub-network has the potential to produce notable regional orbit errors.

Annual difference signals with respect to the REF orbit are most prominent for the GSFC and GRGS solutions, while they are negligible for the SLR, DORIS, and ITRF14 orbits. The corresponding patterns of the annual amplitudes for the differences of REF versus GSFC, GRGS, and Geoid orbits and of GRGS versus GSFC orbits are shown in Fig. 5. The observed patterns for the GSFC and GRGS orbit differences consist of a dipole with centers in the southeastern Indian Ocean and the Caribbean. Since the two centres are phase shifted by half a year, the effect on the global mean differences is marginal. The pattern coincides with the patterns already shown to be related to the use of AOD1B products (Rudenko et al., 2016) and different Earth's time variable gravity fields for TOPEX/Poseidon POD (Esselborn et al., 2016). However, the annual differences between the REF and Geoid orbits can only explain part of the observed differences between the REF and GRGS orbits. In addition, the annual differences between GRGS and GSFC orbits are quite small and show no distinct pattern. Another plausible source of the relatively strong signal for the GSFC and GRGS orbit cases are the differences in the annual corrections for station coordinates by geocenter motion corrections and non-tidal atmospheric loading. A careful consideration of the relevant models used for the POD of these three orbits suggests that the observed differences originate in part from the non-tidal atmospheric loading effect on the stations which was applied for the GFZ but not the GRGS and GSFC orbits. There is evidence that the annual signal from the EIGEN-6S2 gravity field model is closer to the gravity field solution applied for the GSFC orbits than to the one from EIGEN-6S4 – at least in the pre-GRACE period (Fig. 5).

The patterns of the interannual variability of the regional trends are shown in Fig. 6 for all orbit differences. The trend errors reach up to 1.2 (0.9) mm/year for the GSFC (GRGS) orbit differences (Table 5). The patterns of the trend variability from the GSFC and GRGS differences show coinciding maxima in the regions around South America and Australia. The

differences for the Geoid orbit show similar features even though the absolute trend variability is smaller (up to 0.4 mm/year). For the SLR and DORIS orbit differences, the patterns of interannual variability (Fig. 6) are patchy and oriented along individual tracks. For the ITRF14 solution, the trend variability is slightly increased at high latitudes (up to 0.2 mm/year). The patterns of the interannual trend variability derived from the GFZ test orbits suggest that differences in the TVG modelling and contributions from the tracking systems are the most plausible sources of the observed regional differences of trend variability between REF, GSFC, and GRGS orbits.

The strongest regional changes in the decadal trend (Fig. 7 and Table 5) are observed for the differences between the REF and GSFC orbits (up to 1.0 mm/year). For the GSFC orbit, high absolute decadal trend differences tend to coincide with maximum seasonal differences, but not with maximum interannual variability. The differences between the REF and GRGS orbit trends reach 0.7 mm/year at maximum and the patterns of maximum annual amplitudes, interannual and decadal trend differences coincide. The differences between the REF and Geoid orbit trends resemble these patterns. However, the trend values are smaller (up to 0.4 mm/year) and can explain only about half of the observed decadal trend differences. The decadal trends related to EIGEN-6S2 and EIGEN-6S4 differences during the TOPEX period presumably originate from the modelling of the TVG after August 2002, since before drift terms are only applied to degree 2 terms. The degree 2 terms, in turn, are defined by SLR data and show close agreement between the two TVG models for the pre-GRACE period. The ITRF14 orbit differences drift locally by a rate of up to 0.2 mm/year with positive values in the southern hemisphere and negative values in the northern hemisphere, indicating a drift in the z-component between the reference system realisations. The observed values are in good agreement with the combined change of scale and rate of the z-component of the transformation between ITRF2008 and ITRF2014 (Altamimi et al., 2016). The regional decadal trends for the SLR and DORIS orbit differences are patchy and rather related to particular tracks without consistent long-wavelength behaviour. Higher trends of up to 0.4 mm/year emerge for the DORIS orbit. The patterns of the decadal trend differences derived from the GFZ test orbits suggest that differences in the TVG modelling are the most plausible source of the observed regional decadal trend differences between REF, GSFC, and GRGS orbits.

## 3.4 Differences between ascending and descending passes

The crossover point analysis (Table 3) reveals considerable global mean differences between ascending and descending passes for most orbits. Fu and Haines (2013) showed that orbit errors might induce diverging drifts for sea level derived from ascending and descending passes. In the following, we study whether there are systematic changes to the results obtained so far when ascending and descending passes are investigated separately. Therefore, for a subset of orbit solutions the same analyses were performed as before, but for data sets derived from ascending and descending passes only. Since the DORIS orbit reveals the most pronounced median ascending/descending differences we have chosen to study the *REF minus DORIS* and the *REF minus TBias* orbit differences further. During the POD of the GRGS orbit, the DORIS system time bias has not been estimated, therefore, we include the GRGS orbit in the analysis as well. However, in contrast to the previous analysis, we study the difference *Geoid minus GRGS* instead of *REF minus GRGS* in order to exclude the effects of different time variable gravity fields from the analysis.

The global mean radial orbit differences for ascending and descending passes are for all three cases in the range of ±12 mm (Fig. S3). The ascending and descending radial orbit differences are significantly anti-correlated. The correlation coefficient is almost -1 for the *REF minus TBias* case, almost -0.8 for the *Geoid minus GRGS* case, and still -0.5 for the *REF minus DORIS* case. The correlation is further increased for periods of more than one year. The *REF minus TBias* global mean time series resembles the DORIS system time bias applied for the REF orbit (Fig. S1). The global mean radial differences for the *Geoid minus GRGS* case reveal similar features as well. All three orbit differences exhibit diverging global mean radial differences for ascending and descending tracks after the year 2000. The interannual trend variability and decadal trends derived from the analysis of the global mean radial orbit difference series over the oceans are summarized in Table 6 for the merged, ascending, and descending passes. If the ascending and descending passes are analysed separately, the interannual trend variability is increased by at least five times for the corresponding orbit differences. Ascending passes exhibit higher variability than the descending. The differences for the global mean decadal trends between ascending and descending passes are a multiple of the values for the merged data and reach up to 0.6 mm/year, where both data sets are drifting in opposite directions.

The regional patterns of the decadal trend differences for ascending and descending passes are shown in Fig. 8. The DORIS orbit differences reveal a striking spread between the decadal trends of the ascending and descending passes. The trends are opposite for ascending and descending passes for most areas of the global ocean and reach regionally absolute values of up to 0.8 mm/year. Trends for the *REF minus TBias* orbit differences are very similar but smaller than the *REF minus DORIS* orbit ones. The corresponding analysis for the *Geoid minus GRGS* orbit differences shows again very similar features as for the DORIS differences. This indicates that discrepancies in the reference systems of the tracking stations (distribution of tracking stations, observation sampling, etc.) might give rise to long-wavelength orbit errors being anti-correlated for ascending and descending passes. Relevant contributions are originating from uncertainties of the timing of the DORIS measurements. Increasing time biases are related to increasing along-track position errors and seem to be transferred to radial orbit errors. This mechanism is not fully understood but a further analysis is beyond the scope of this paper. The

350 uncertainties are especially pronounced in tropical and subtropical regions. On regional scales, the interannual and decadal trend errors derived from ascending/descending passes separately can be many times higher than the values derived from the merged data. Even though such effects tend to cancel, whenever both components are merged, they might still introduce considerable errors in regional studies, that are based on along-track data, e.g. at calibration sites.

### 3.5 Regional orbit errors and sea level variability

355 Our analysis exhibits large scale patterns of the orbit related error. Errors for interannual to decadal sea level trends of more than 1 mm/year might hamper the interpretation of the observed sea level variability from altimetry, at least apart from the large oceanic currents. In order to define regions where the orbit related error should be considered when analysing sea level data from TOPEX, we have determined areas with orbit errors of at least 10 % of the corresponding sea level value. Fig. 9 shows the sea level variability, seasonal signal, interannual and decadal trends derived from the ESA CCI TOPEX altimeter 360 data for those regions where the orbit error amounts to at least 10 % of the corresponding sea level value. Taking into account the total orbit related error, about half the ocean is affected. This includes especially calm oceanic regions, whereas for energetic regions like the Circumpolar Current, Tropical Pacific, and the western boundary currents of the northern hemisphere the dynamic ocean signal is much larger than the orbit error. For the seasonal signal, mainly the Southern Ocean is concerned. Critical regions for the estimation of the interannual variability are the Tropical and Subtropical Atlantic and 365 the south-eastern Pacific. For decadal scales, the orbit related trend errors are prominent in a couple of regions including: South Atlantic, western North Atlantic, central Pacific, and south-eastern Indian Ocean, but also several marginal seas including the Mediterranean, Red Sea, Yellow Sea and Sea of Japan.

### 4 Summary and Conclusions

We have investigated the radial orbit error budget associated with three state-of-the-art orbit solutions from GFZ, GSFC and 370 GRGS over the first altimetry decade (1993-2004). It is crucial to know the accuracy of these early altimeter data in order to judge the reliability of long-term sea level trends and of estimates of the acceleration of global mean sea level rise. For this purpose, we have chosen the TOPEX/Poseidon mission, since it is the reference altimetry mission used in the ESA CCI Sea Level project over this time span. We estimate the orbit errors from the radial orbit differences which implies that errors common to all orbits can not be detected. However, since the three orbits were derived using various up-to-date models, the 375 errors common to the three orbits should be rather low which makes us confident that our error estimates represent most of the error. A set of five test orbit solutions derived at GFZ is used to estimate the contributions of the most significant factors to the error budget. We have focused on the impact of uncertainties of the tracking station sub-networks (SLR and DORIS), of DORIS system time bias, of the reference frame, and of the Earth's time variable gravity field models on the radial orbit component, and hence the derived sea level. The estimates of the radial orbit errors at seasonal, interannual (5 years), and 380 decadal time scales are given in Table 4 for the global mean sea level and in Table 5 for the regional sea level.

According to our study, the contribution of orbit uncertainties to the error of the global mean sea level during the TOPEX period are of the order of 1.2 mm, which corresponds to more than 10 % of the variability of the global mean sea level (10 mm). The global mean annual (seasonal) component of the radial error is well below 1 mm and can be neglected. The orbit related errors of the decadal trends are up to 0.08 mm/year and should not induce any significant artificial global mean sea level trends. However, on time scales of five years the trend variability may reach up to 0.1 mm/year, which amounts to almost 20 % of the corresponding sea level variability (0.55 mm/year), and could potentially hamper the detection of sea level acceleration from the altimeter data. The major contributions to this error (0.04 – 0.11 mm/year) are, most probably, discrepancies of the station sub-networks (DORIS or SLR) used. The contributions of Earth's time variable gravity field model and the ITRF realisation (ITRF2008 versus ITRF2014) to the global mean error are of only minor importance (0.03 mm/year). These values are in line with the mean upper bound orbit errors given by Couhert et al. (2015) derived for Jason-1 and Jason-2 orbits for the second altimetry decade (2002-2012).

For regional scales, the maximum RMS of the gridded radial orbit error is more than 10 mm. This error is notably less than the 3-4 cm radial orbit error obtained for TOPEX/Poseidon by Marshall et al. (1995) indicating the advance in orbit modelling for this satellite over the past 20 years. However, this error includes a large fraction of sub-seasonal variability which is not subject of this study. The regional upper bound error of the seasonal signal is 6 mm, of the interannual trend variability 1.2 mm/year, and of the decadal trend 1 mm/year. Errors for interannual to decadal sea level trends of more than 1 mm/year might hamper the interpretation of the observed sea level variability from altimetry. For about half of the ocean outside the energetic regions (e.g. Circumpolar Current, Tropical Pacific, Gulf Stream and Kuroshio System) the orbit related errors reach at least 10% of the observed sea level variability. For the seasonal signal, mainly the Southern Ocean is concerned. Critical regions for the estimation of the interannual variability are the Tropical and Subtropical Atlantic and the south-eastern Pacific. For decadal scales, the orbit related trend errors are prominent in a couple of regions including: South Atlantic, western North Atlantic, central Pacific, and south-eastern Indian Ocean, but also several marginal seas including the Mediterranean, Red Sea, Yellow Sea and Sea of Japan.

When using ascending and descending passes separately, the interannual and decadal trend errors can reach multiples of the values derived from the merged data. This is the case for global mean values as well as for regional values. The corresponding large scale pattern is coherent for low and medium latitudes and is strongly anti-correlated for ascending and descending passes. Even though such effects tend to cancel, whenever both components are merged, they might still introduce considerable errors in regional studies, that are based on along-track data, e.g. at calibration sites.

Orbit errors related to discrepancies between the tracking station sub-networks (distribution of tracking stations, observation sampling, etc.) are studied based on GFZ's SLR, DORIS, and TBias orbit solutions. Using SLR and DORIS observations for TOPEX POD together reduces (improves) the RMS of the altimetry single-satellite crossover differences considerably (2-3%), though the DORIS observations seem to aggravate the median differences between ascending and descending passes. The proper estimation of the DORIS system time bias has proven to be a critical factor for the minimization of this effect. The most significant changes are observed for the DORIS orbit solution suggesting that uncertainties of the SLR station sub-

network should have the most prominent effects on the orbit accuracy – at least for GFZ's orbit solutions. This fact is, most probably, related to the weighting factors applied to the observations within the GFZ orbit determination process. Using the latest reference frame ITRF2014 instead of the predecessor ITRF2008 slightly improves the accuracy of the TOPEX/Poseidon orbit solution. The contribution of the uncertainties in the ITRF realisation to the regional upper bound error is only marginal. Errors induced by uncertainties of the Earth's time variable gravity field model are studied on the base of GFZ's Geoid orbit solution. The orbit evaluations show that the Geoid orbit performs slightly better than the REF orbit in the pre-GRACE period due to differences in the periodic annual and semiannual variations applied to the TVG field models. Uncertainties of the gravity field model give rise to orbit errors at all analysed periods. We estimate regional upper bound errors of ~3 mm for the seasonal signal and of 0.4 mm/year for the interannual trend variability and the decadal trend. This accounts for about 60 % of the seasonal, about 30 % of the interannual, and about 40 % of the decadal orbit error which are related to differences between EIGEN-6S2 and EIGEN-6S4.

The regional upper bound radial orbit errors obtained from our study are by factor 2 to 5 smaller than the ones reported by Couhert et al. (2015) for the period 2002 to 2012. This might partly reflect recent improvements of the stability of reference frames which result in smaller changes from ITRF2008 to ITRF2014 than previously from ITRF2005 to ITRF2008. However, the accuracy of the Earth's time variable gravity model and the tracking observations for the 1990's should be inferior to more recent periods. The error related to the uncertainties of the tracking station sub-networks might be underrated in our study since all analysed orbits rely on basically the same set of tracking observations. The effect of uncertainties of the time variable gravity (TVG) field might be underestimated as well, since both EIGEN-6S4 and EIGEN-6S2 model the TVG field in the pre-GRACE period by periodic annual and semi-annual variations derived from GRACE plus annual values and drift terms for degree two terms derived from SLR measurements. In contrast, the TVG field used for the GSFC orbit determination is weekly changing. Using SLR measurements to geodetic cannon-ball satellites (Sośnica et al., 2015, Bloßfeld et al., 2016) and in combination with DORIS measurements to altimetry and remote sensing satellites (Lemoine et al., 2016) allows to determine Earth's time variable gravity for the period 1993-2003, i.e. before GRACE, more precisely than just using SLR measurements to LAGEOS-1/2. Combined use of SLR and DORIS measurements to numerous geodetic satellites, especially for 1990-2003, with the GRACE measurements should further improve Earth's time variable gravity field models and hence further enhance orbit solutions for the ERS and the TOPEX/Poseidon altimetry missions.

**Acknowledgements**

We are grateful for the insightful comments of the reviewer Dr. Nikita Zelensky which helped to improve the manuscript substantially. We thank Goddard Space Flight Centre and Groupe de Recherche de Geodesie Spatiale for providing GSFC std1504 and GRGS orbit solutions, ESA CCI for along-track TOPEX Sea Level v1.1 ECV data, and Joana Fernandes for providing updated wet troposphere corrections (GPD+). This research was partly supported by the European Space Agency (ESA) within the Climate Change Initiative Sea Level (SLCCI) Phase II project and by the Deutsche

Forschungsgemeinschaft (DFG) through grant CoRSEA as part of the Special Priority Program (SPP)-1889 "Regional Sea Level Change and Society" (SeaLevel) and within the projects "Consistent dynamic satellite reference frames and terrestrial geodetic datum parameters" and "Interactions of low-orbiting satellites with the surrounding ionosphere and thermosphere (INSIGHT)" and by the International Office of the BMBF under the grant 01DO17017 "Sea Level Change and its Hazardous Potential in the East China Sea and Adjacent Waters" (SEAHAP).

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

Table 1. The main models used for calculation of GFZ VER11, GSFC std1504 and GRGS orbits

| Parameter | GFZ REF (VER11) orbit | GSFC std1504 orbit | GRGS orbit |
|---|---|---|---|
| Terrestrial reference frame | ITRF2008 (Altamimi et al., 2011), SLRF2008 (Pavlis 2009), DPOD2008 (Willis et al., 2016) | ITRF2008, SLRF2008, DPOD2008 | ITRF2008, SLRF2008, DPOD2008 |
| Polar motion and UT1 | IERS EOP 08 C04 (IAU2000A) series with IERS diurnal and semi-diurnal variations | IERS Bulletin A daily (consistent with ITRF2008), diurnal and semi-diurnal variations | IERS EOP 08 C04 |
| Precession and nutation model | IERS Conventions (2010) | IAU2000 | IERS 2010 using non-rotating origin |
| Station displacements due to annual geocenter variations | None | Ries (2013) | None |
| Non-tidal atmospheric loading effect on stations | Based on ECMWF ERA-Interim data | None | None |
| Ocean loading effect on stations | FES2004 (Lyard at al., 2006) | GOT4.10 (Ray, 2013) | FES2012 (Carrère et al., 2012) |
| Static Earth's gravity field model | EIGEN-6S4 (Förste et al., 2016) degree/order 81-90 | GOCO2S (> d/o5) (Goiginger et al., 2011) | EIGEN-6S2 (Rudenko et al., 2014) |
| Time-variable Earth's gravity field model | EIGEN-6S4 degree 2: yearly value and drift term, d/o 1-80: periodic (semi-) annual variations, from 15.8.2002: yearly values, drift terms and (semi-)annual variations for d/o 1-80. | Updated harmonic piece-wise fit weekly solutions (Lemoine et al., 2016) up to d/o 5 | EIGEN-6S2 degree 2: yearly value and drift term, d/o 2-50: periodic (semi-) annual variations, from 1.1.2003: yearly values and drift terms for d/o 2-50 |
| Solid Earth tide | IERS Conventions (2010) | IERS Conventions (2003) | IERS Conventions (2010) |
| Ocean tide model | EOT11a (Savchenko and Bosch, 2012) up to d/o 80 | GOT4.10 up to d/o 50 | FES2012 up to d/o 50 |
| Non-tidal atmospheric and oceanic gravity | GFZ AOD1B RL05 up to d/o 100 (Dobslaw et al, 2013), including ECMWF 6-hourly fields and OMCT | ECMWF 6-hourly fields up to d/o 50 | 3-hourly ERA-interim and TUGO R12 up to d/o 50 |

| | | | |
|---|---|---|---|
| Atmospheric density model | MSIS-86 (Hedin, 1987) | MSIS-86 | DTM 94, with best available solar activity data |
| Earth radiation and albedo | Knocke et al. (1988) | Knocke et al. (1988) | Albedo and IR pressure values interpolated from ECMWF 6hr grids |
| Radiation pressure model | Tuned 8-panel (Cerri and Ferrage, 2016) | Tuned 8-panel | Thermo-optical coefficient from pre-launch box and wing model, with smoothed Earth shadow model |
| Tracking data | SLR, DORIS | SLR, DORIS | SLR, DORIS |
| SLR tropospheric correction model | Mendes and Pavlis (2004) | Mendes and Pavlis (2004) | Mendes and Pavlis (2004) |
| DORIS tropospheric correction model | Vienna Mapping Functions 1 (Boehm and Schuh, 2004) | Vienna Mapping Functions 1 | GPT2/Vienna Mapping Functions 1 |
| DORIS modelling | DORIS beacon frequency bias modelling | DORIS beacon phase center | DORIS beacon phase center |
| DORIS system time bias | Estimated once per arc | Estimated once per arc | None |
| Drag coefficients | Estimated every 6 h | Estimated every 8 h | Estimated every 12 h |
| Along- and cross-track empirical accelerations (once per revolution) | Estimated every 24 h | Estimated every 24 h | Estimated once per arc (3.5 days) |
| SLR antenna reference | LRA model (note 1 below) | LRA model (note 1 below) | X: 1.2429, Y: -0.0012, Z: 0.8783 in [m] |
| DORIS antenna reference | pre-launch | pre-launch | pre-launch |
| SLR / DORIS observation weight | 3 cm / 0.05 cm/s | 10 cm / 0.2 cm/s | 1 cm / 0.03 cm/s |

Note 1: https://ilrs.cddis.eosdis.nasa.gov/missions/satellite_missions/past_missions/topx_com.html



**Table 2. Average values of SLR and DORIS RMS fits, radial, cross-track and along-track two-day arc overlaps and the number of the arcs used to compute these values for the reference and five test orbits.**

| Orbit name | SLR RMS [cm] | DORIS RMS [cm/s] | Radial arc overlap [cm] | Cross-track arc overlap [cm] | Along-track arc overlap [cm] | Number of arcs used for SLR RMS | Number of arcs used for DORIS RMS | Number of arc overlaps used | Comment on the orbit |
|---|---|---|---|---|---|---|---|---|---|
| REF | 1.96 | 0.04778 | 0.90 | 6.52 | 3.65 | 494 | 459 | 433 | Reference |
| SLR | 1.59 | — | 1.72 | 7.23 | 9.54 | 494 | — | 425 | SLR only |
| DORIS | — | 0.04795 | 0.88 | 6.84 | 2.96 | — | 459 | 392 | DORIS only |
| TBias | 1.99 | 0.04785 | 0.85 | 6.45 | 2.78 | 494 | 459 | 433 | No DORIS system time bias estimated |
| ITRF14 | 1.97 | 0.04776 | 0.84 | 6.45 | 2.83 | 494 | 459 | 433 | ITRF2014 |
| Geoid | 1.96 | 0.04775 | 0.83 | 6.43 | 2.80 | 494 | 459 | 433 | EIGEN-6S2 |

**Table 3: Crossover point analysis: median and RMS values of global mean height differences for maximum time lapses of 5 days for all orbit solutions during the period April 1993 — September 2004. The highest and lowest values of each quantity are marked bold.**

| Crossover differences | REF | GSFC | GRGS | SLR | DORIS | TBias | ITRF14 | Geoid |
|---|---|---|---|---|---|---|---|---|
| Median [mm] | -3.1 | **-1.6** | -3.0 | -2.7 | **-4.7** | -3.6 | -2.8 | -2.1 |
| RMS [mm] | 49.8 | **49.5** | **51.3** | 51.2 | 50.7 | 49.8 | 49.8 | 49.7 |

**Table 4: Estimates of global mean orbit related errors for the total signal, interannual , and decadal trend. Values are derived from the mean radial orbit differences over the oceans: *REF minus SLR*, *REF minus DORIS*, *REF minus ITRF14*, *REF minus Geoid*, *REF minus GSFC*, *REF minus GRGS* for the period April 1993 — June 2004. The corresponding values derived from the altimetric sea level anomalies (SLA) are added for comparison. Details on the estimation method are given in Sect. 3.1.**

| Global mean error | REF-SLR | REF-DORIS | REF-ITRF14 | REF-Geoid | REF-GSFC | REF-GRGS | SLA |
|---|---|---|---|---|---|---|---|
| RMS [mm] | 0.7 | 1.8 | 0.2 | 0.3 | 1.1 | 1.2 | 10.4 |
| RMS 5-year trend [mm/year] | 0.04 | 0.11 | 0.03 | 0.02 | 0.07 | 0.10 | 0.55 |
| Δ decadal trend [mm/year] | 0.00 | 0.06 | 0.01 | 0.00 | 0.08 | 0.02 | 2.89 |

**Table 5: Estimates of regional maximum orbit related errors for the total and seasonal signal, nterannual and decadal trend. Values are derived from the radial orbit differences: *REF minus SLR*, *REF minus DORIS*, *REF minus ITRF14*, *REF minus Geoid*, *REF minus GSFC*, *REF minus GRGS* for the periods April 1993 ─ June 2004. Details on the estimation method are given in Sect. 3.1.**

| Regional maximum error | REF-SLR | REF-DORIS | REF-ITRF14 | REF-Geoid | REF-GSFC | REF-GRGS |
|---|---|---|---|---|---|---|
| RMS [mm] | 7.2 | 9.3 | 2.4 | 3.5 | 7.4 | 10.7 |
| Annual amplitude [mm] | 1.4 | 2.1 | 0.4 | 3.2 | 5.4 | 5.6 |
| RMS 5-year trend [mm/year] | 0.5 | 0.6 | 0.2 | 0.4 | 1.2 | 0.9 |
| Δ decadal trend [mm/year] | 0.2 | 0.4 | 0.2 | 0.4 | 1.0 | 0.7 |


**Table 6: Differences of interannual trend variability and decadal trend formerded, ascending, and descending passes related to the orbit solution. Values are derived from the mean radial orbit differences over the oceans: *Geoid minus GRGS*, *REF minus DORIS*, and *REF minus TBias* for the period April 1993 ─ June 2004. Values for ascending and descending passes are given in brackets.**

| Global mean differences | Geoid-GRGS | REF-DORIS | REF-TBias |
|---|---|---|---|
| RMS 5-year trend [mm/year] | 0.10 (0.62, 0.48) | 0.11 (0.53, 0.38) | 0.02 (0.55,0.57) |
| Δ decadal trend [mm/year] | -0.02 (0.30, -0.34) | -0.06 (0.20, -0.27) | -0.01 (0.10,-0.13) |

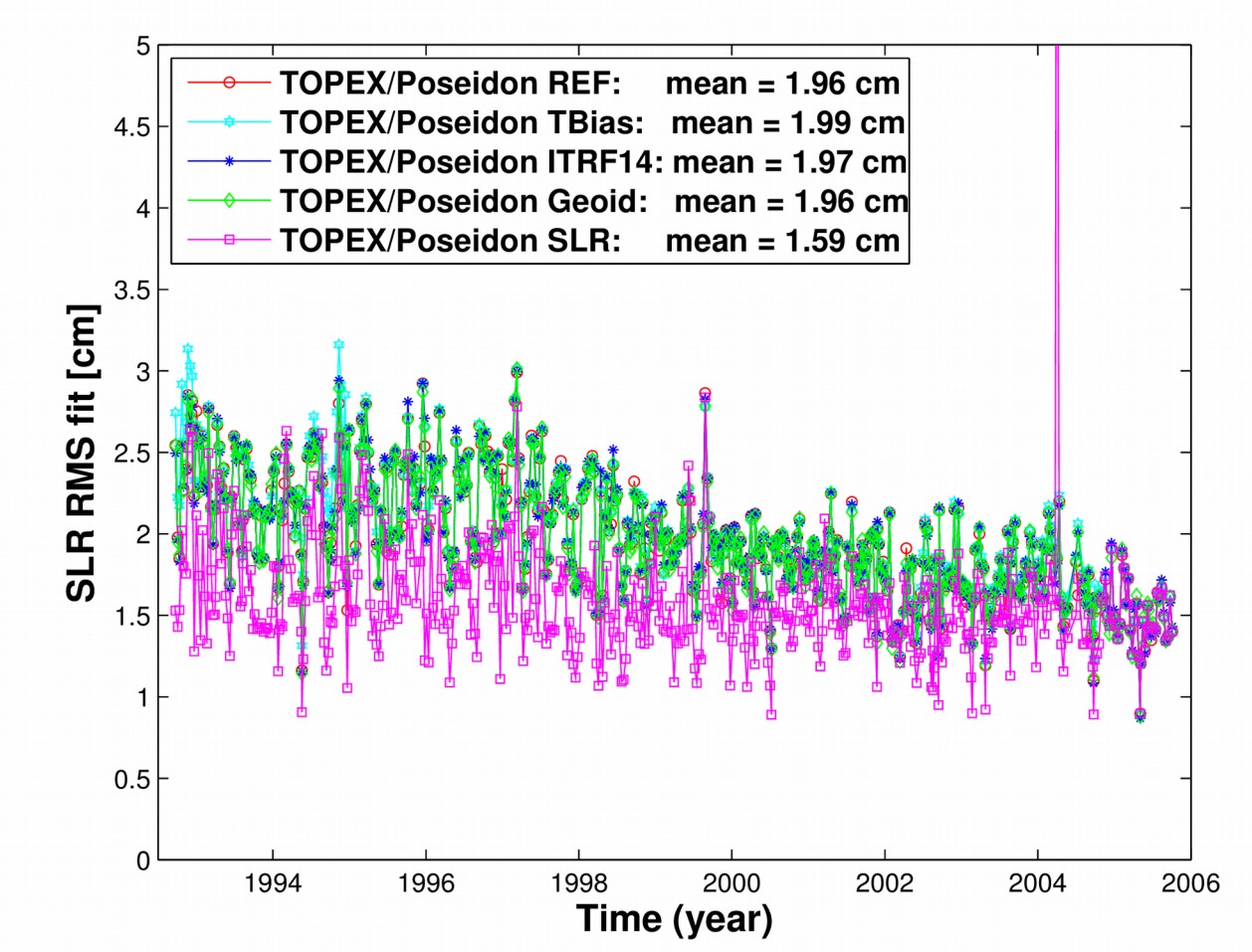

**Figure 1: SLR RMS fits of TOPEX/Poseidon REF, SLR, TBias, ITRF14, and Geoid orbits.**

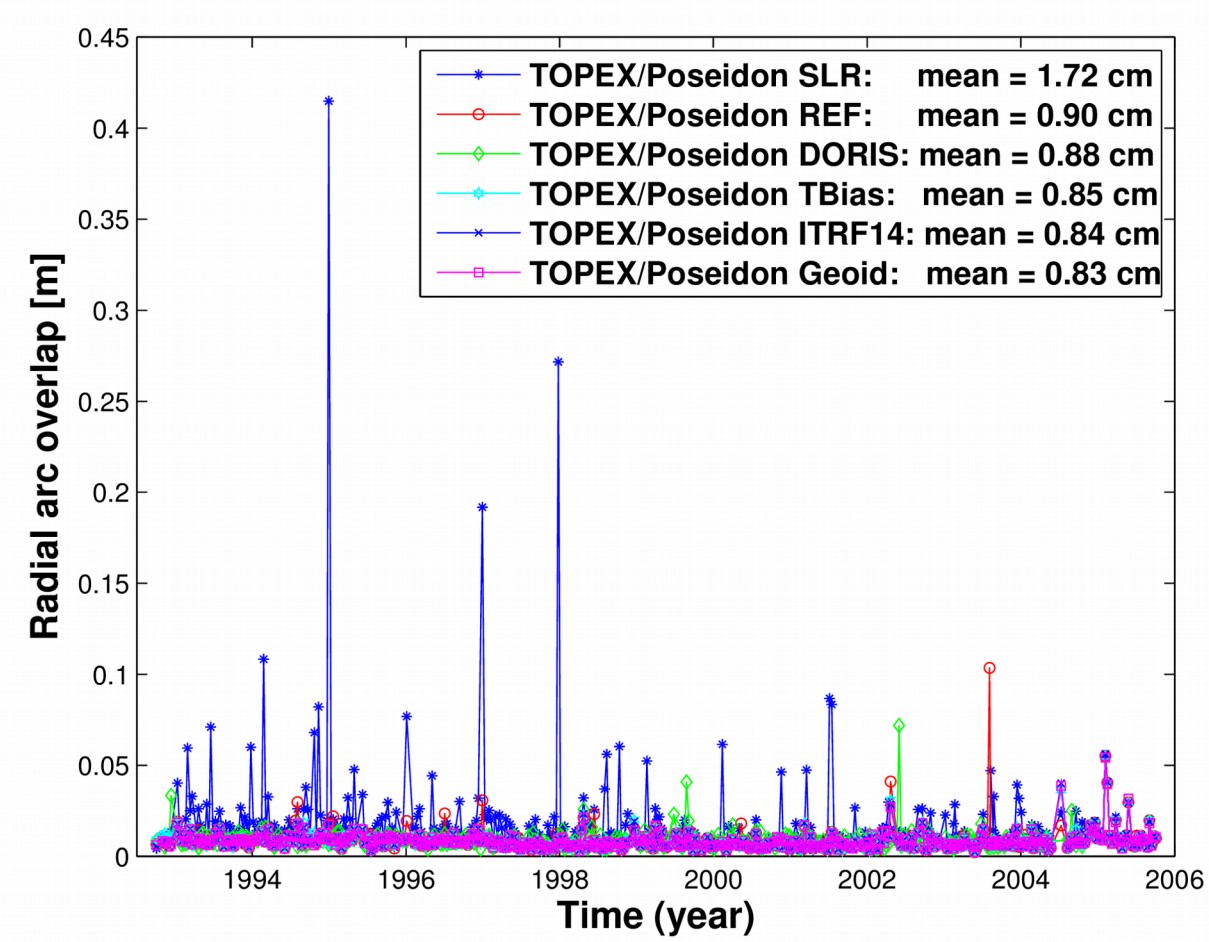

Figure 2. Radial arc overlaps of TOPEX/Poseidon REF, SLR, DORIS, TBias, ITRF14, and Geoid orbits.



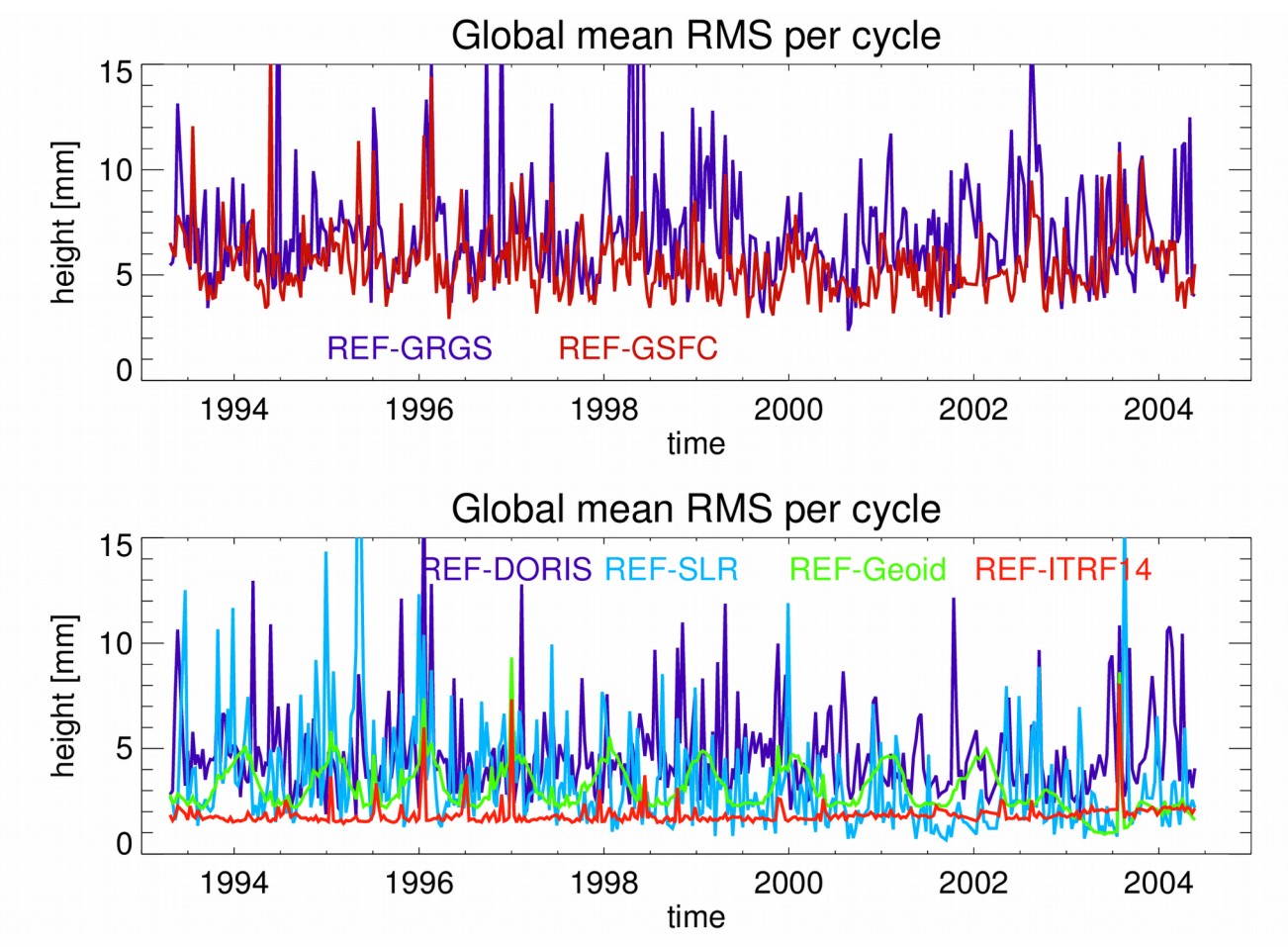

**Figure 3: Time series of the global mean RMS per cycle of gridded radial orbit differences for *REF minus GSFC* (dark blue), and *REF minus GRGS* (red) on the top; for *REF minus DORIS* (dark blue), *REF minus SLR* (light blue), *REF minus Geoid* (green), and *REF minus ITRF14* (red) on the bottom.**



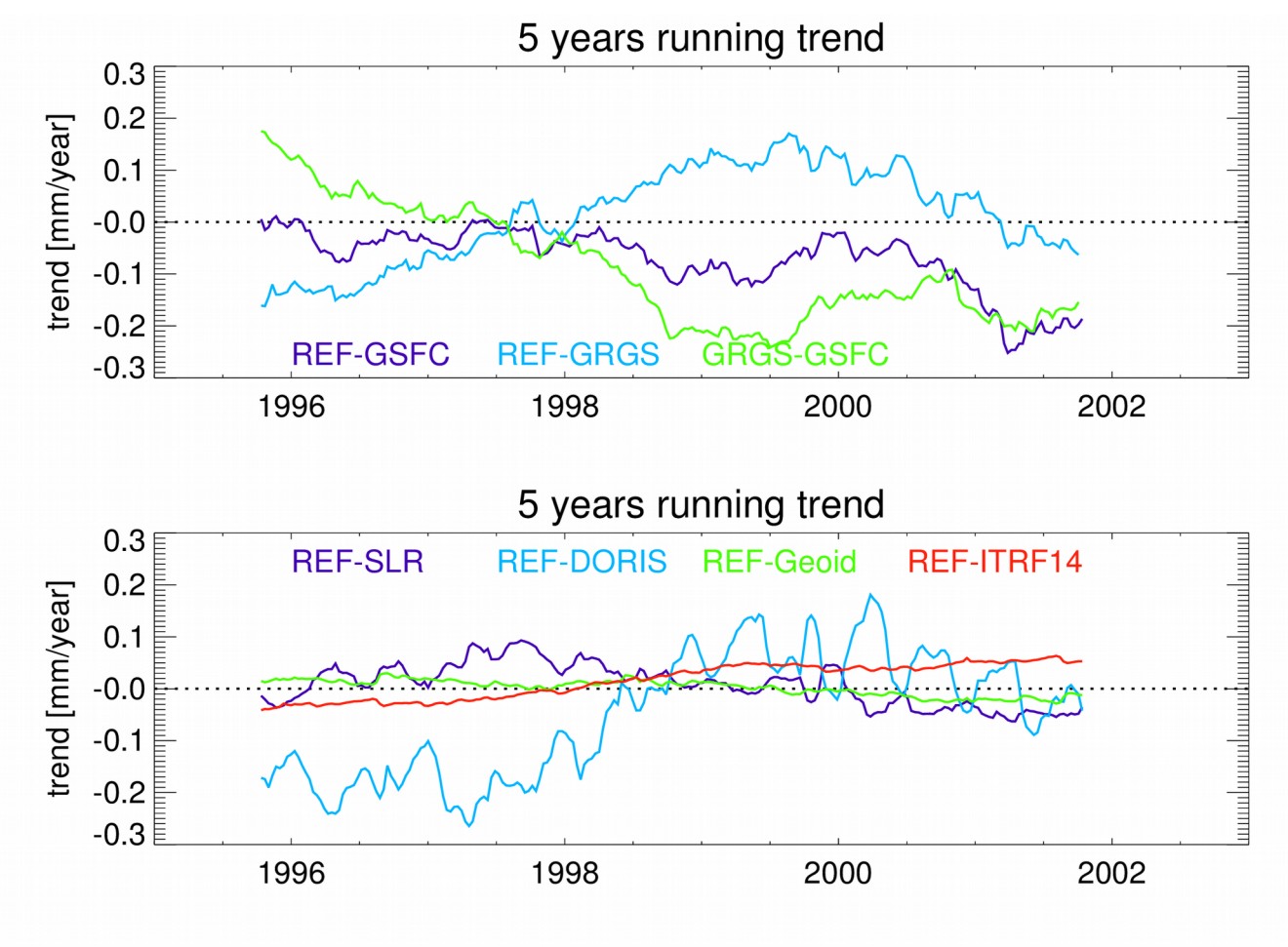

**Figure 4: 5-year running trends for the global mean radial orbit differences over the oceans for *REF minus GSFC* (dark blue),**
**695** *REF minus GRGS* (light blue), and *GRGS minus GSFC* (green) on the top; for *REF minus SLR* (dark blue), *REF minus DORIS*
**(light blue), *REF minus Geoid* (green), and R*EF minus ITRF14* (red) on the bottom. Trend values are given for the central time of**
**the corresponding running window.**


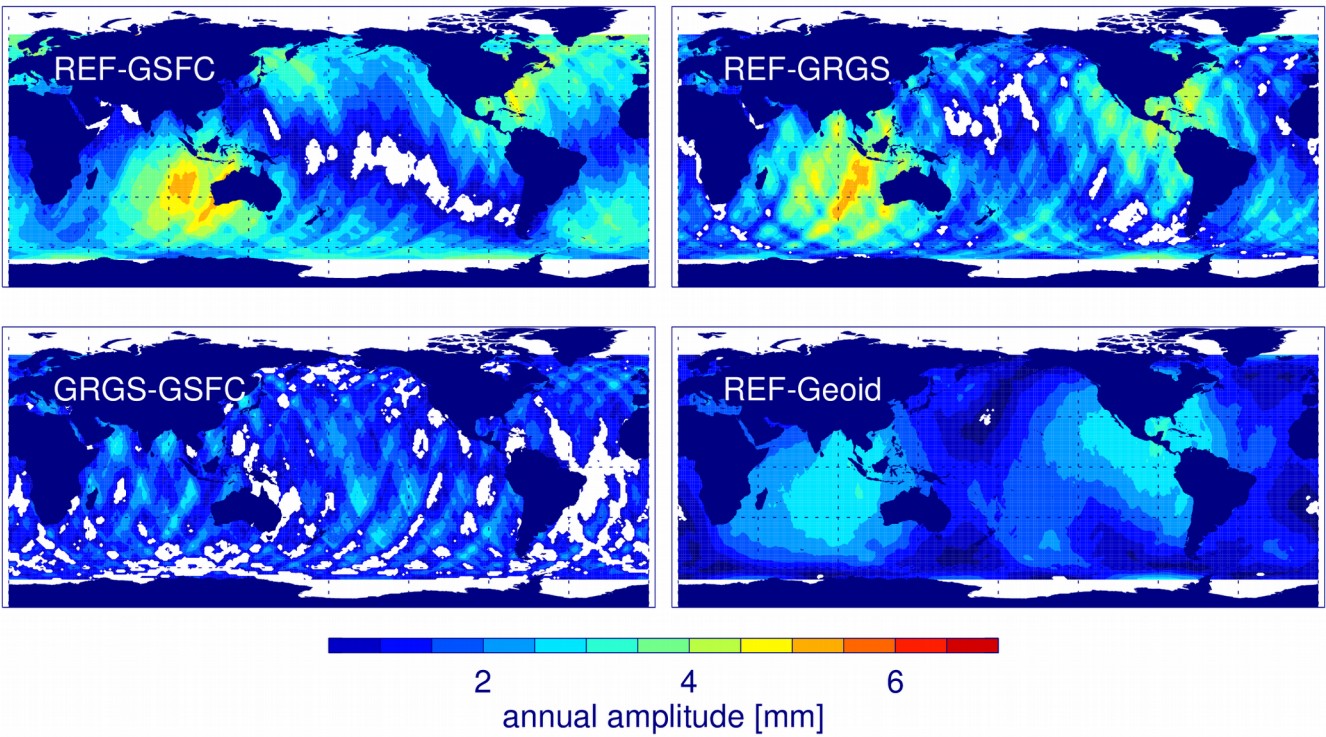

Figure 5: Annual amplitude of the radial orbit differences for *REF minus GSFC*, *REF minus GRGS*, *GRGS minus GSFC*, and *REF minus Geoid*. The regions with formal errors larger than the fitted value are masked out (white). The maximum amplitude difference is given in Table 5.

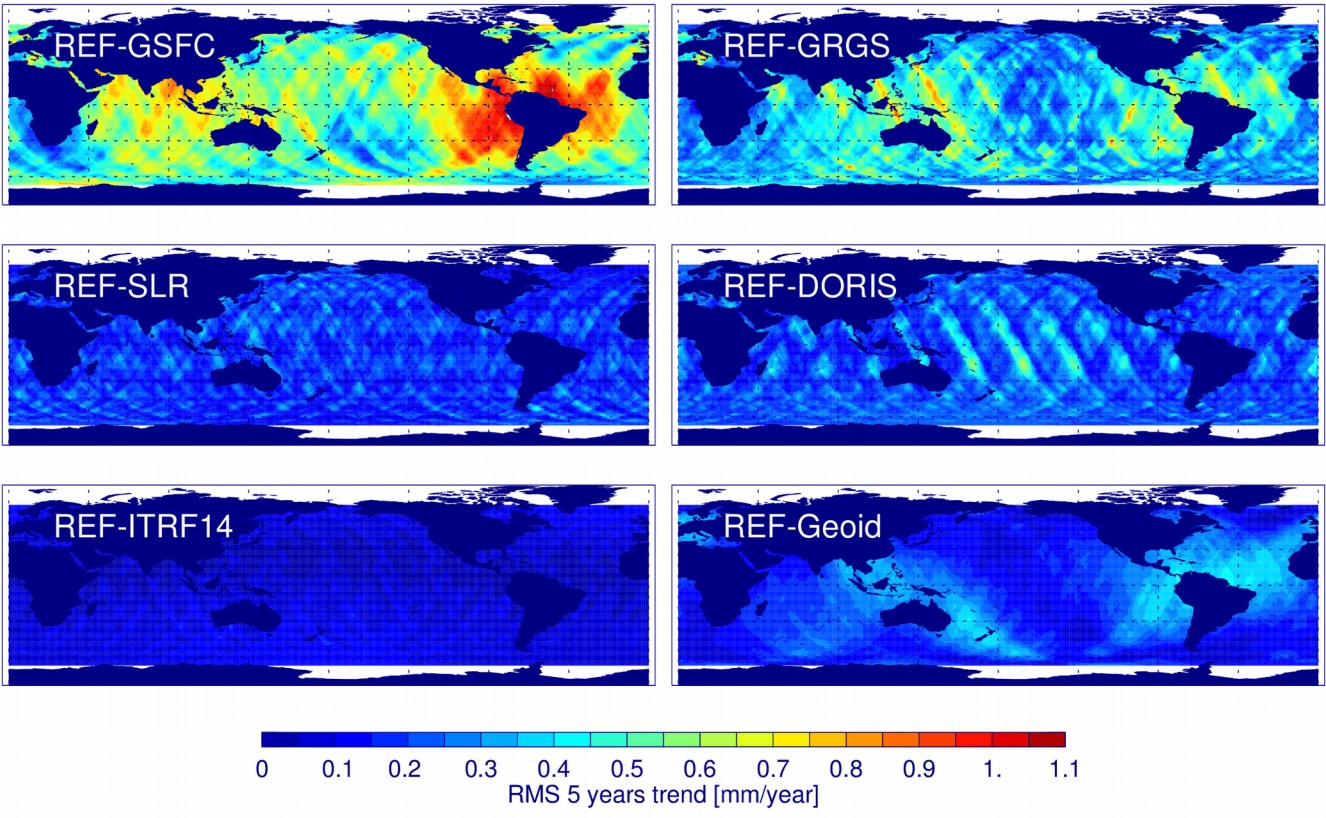

**Figure 6: RMS of 5-year running trend differences of the radial orbit components for *REF minus GSFC*, *REF minus GRGS*, *REF minus SLR*, *REF minus DORIS*, *REF minus ITRF14*, and *REF minus Geoid* for the period April 1993 ─ June 2004. The global mean RMS of the differences over the ocean is given in Table 4.**




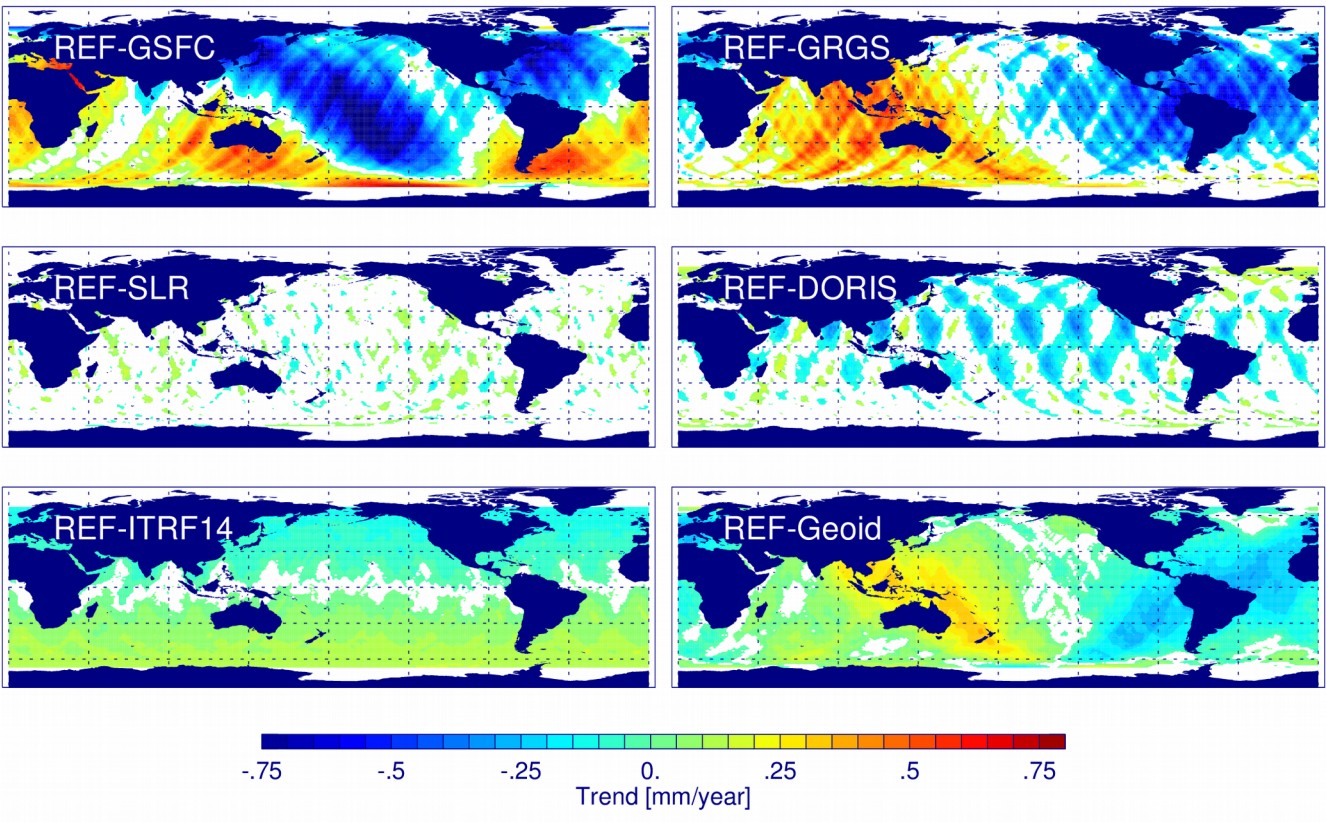

**Figure 7: Decadal trend differences of radial orbit components for *REF minus GSFC*, *REF minus GRGS*, *REF minus SLR*, *REF minus DORIS*, *REF minus ITRF14*, and *REF minus Geoid* for the period April 1993─June 2004. Regions with formal errors larger than the fitted value are masked out (white). The global mean trend difference over the ocean is given in Table 4.**




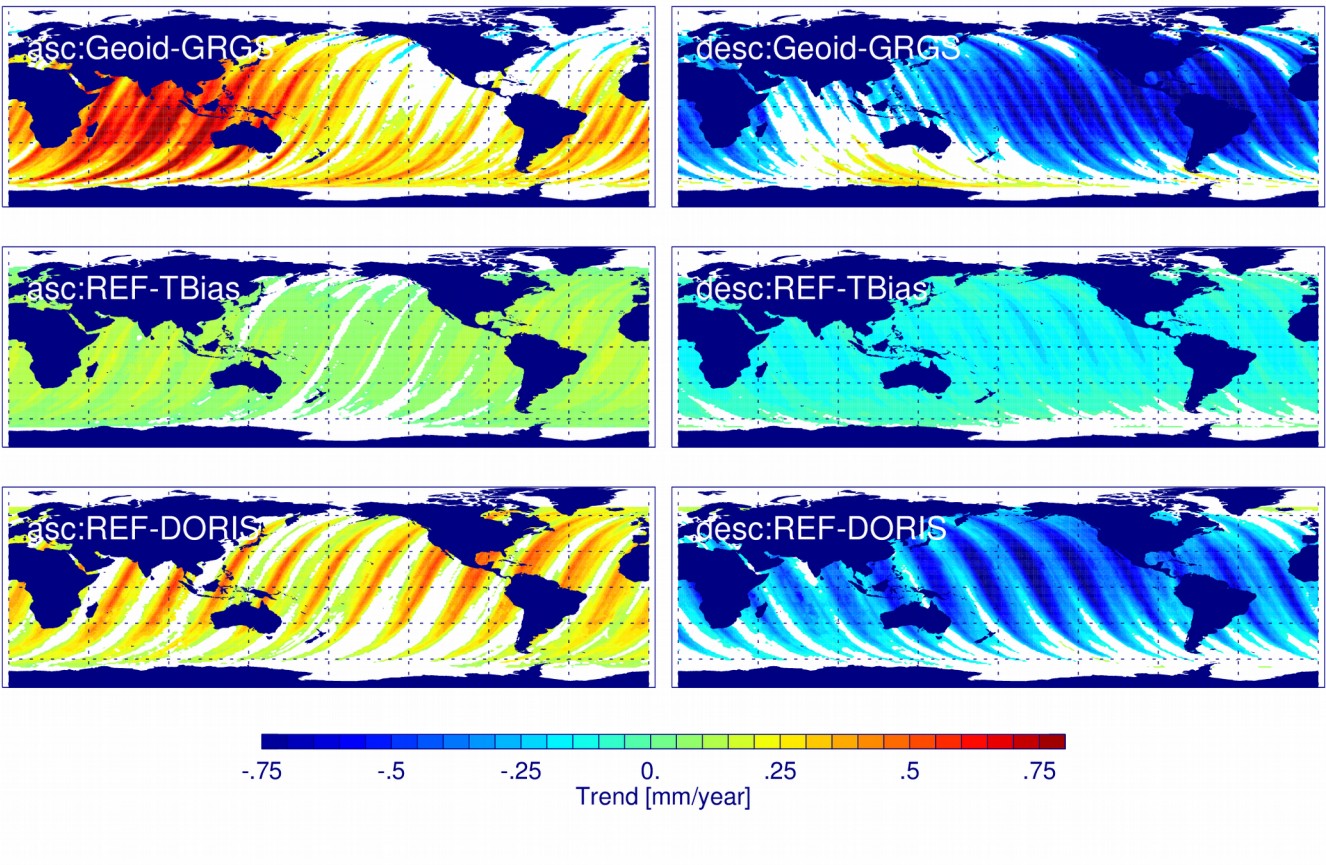

**Figure 8: Decadal trend differences of radial orbit components for ascending (left) and descending (right) passes for *Geoid minus GRGS REF minus TBias*, and *REF minus DORIS* for the period April 1993 ─ June 2004. Regions with formal errors larger than the fitted value are masked out (white). The global mean trend difference over the ocean is given in Table 6.**


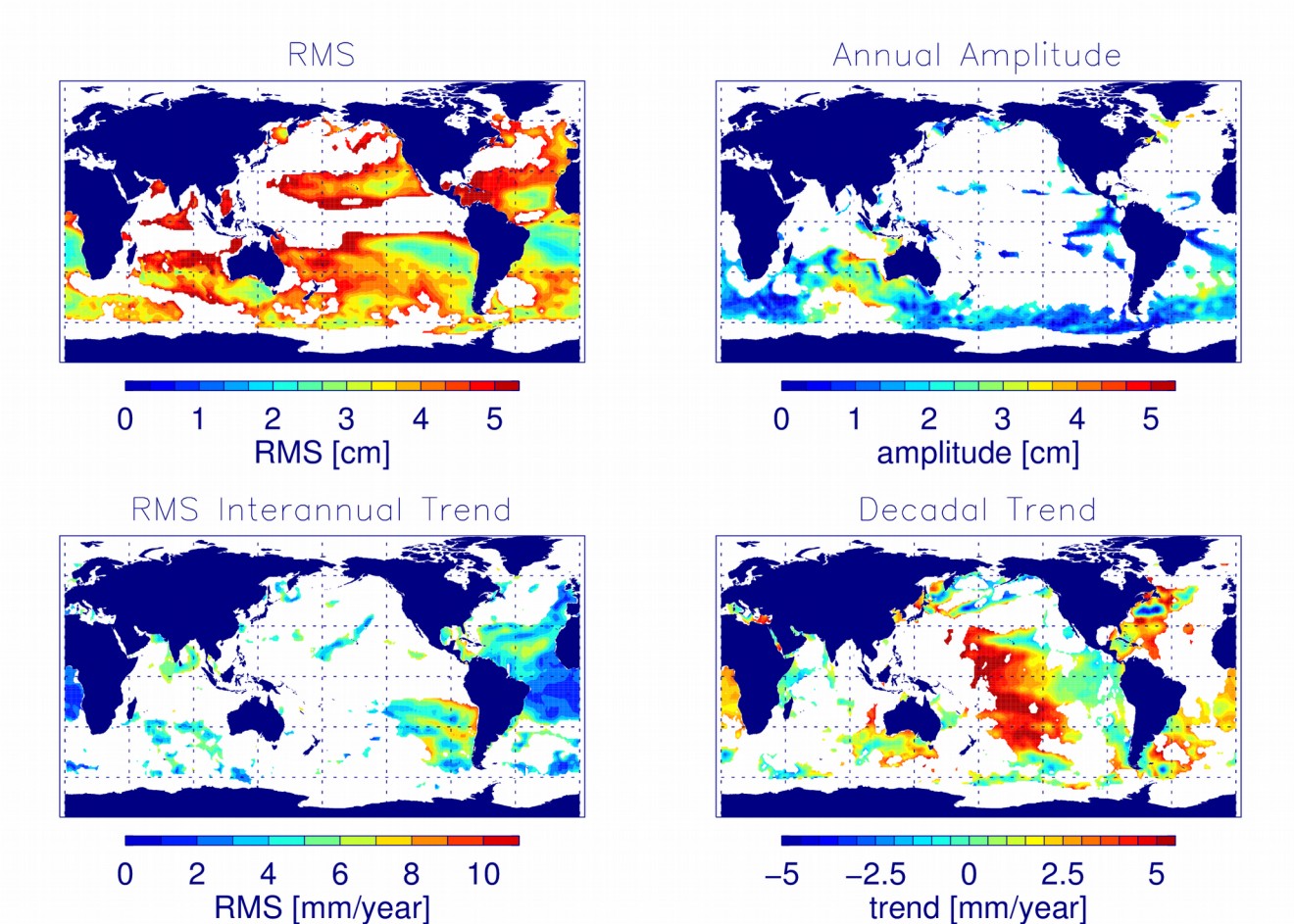

**Figure 9: RMS of sea level, annual amplitude, RMS of interannual (5 years) running trend, and decadal trends from TOPEX altimeter data for the period February 1993─ October 2005. Colour coded are sea level values for which the local orbit errors (estimated from *GFZ minus GRGS)* reach more than 10 % of the local sea level values. All other regions are masked out (white).**


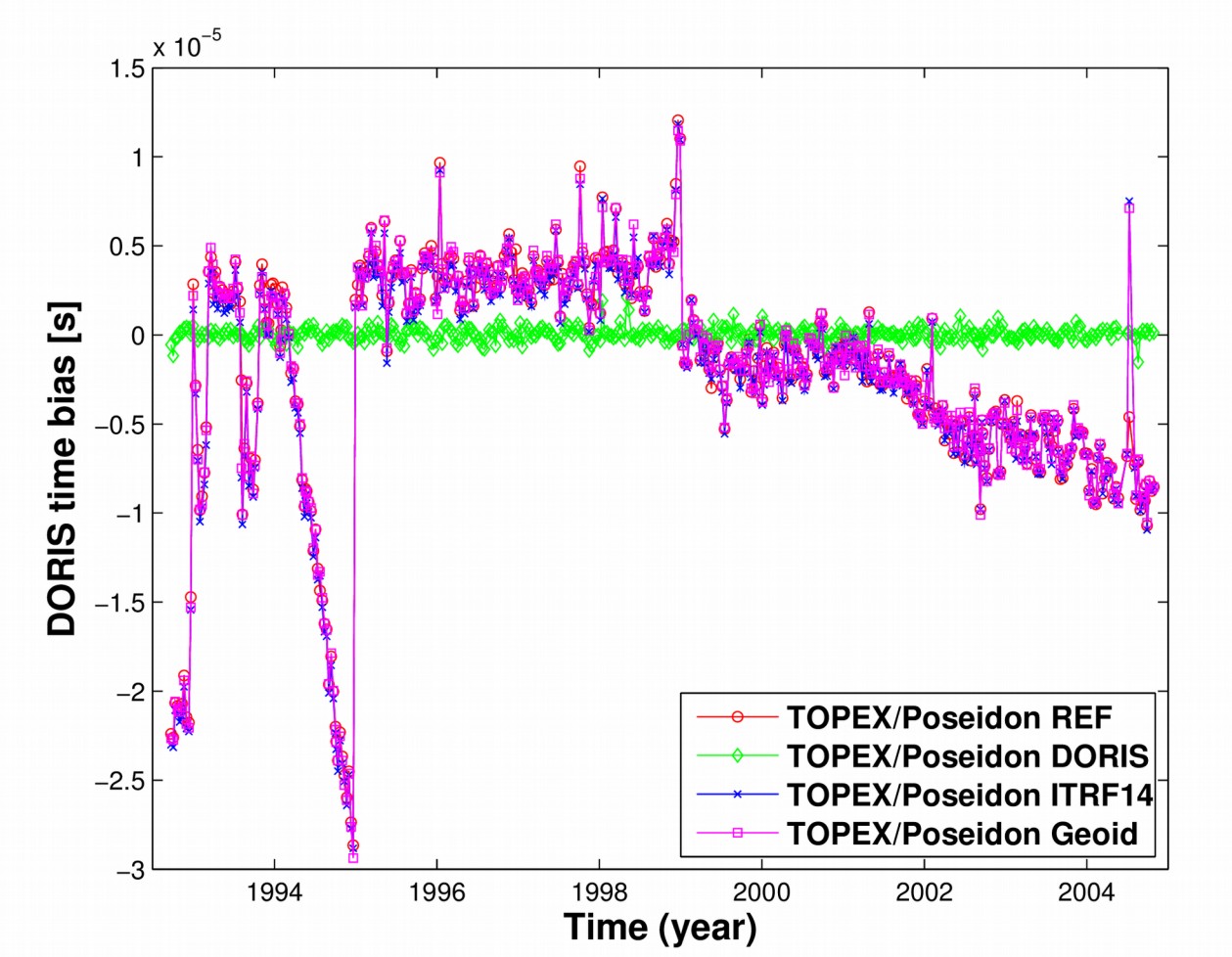

**Figure S1: DORIS system time bias of TOPEX/Poseidon REF, DORIS, ITRF14, and Geoid orbits.**

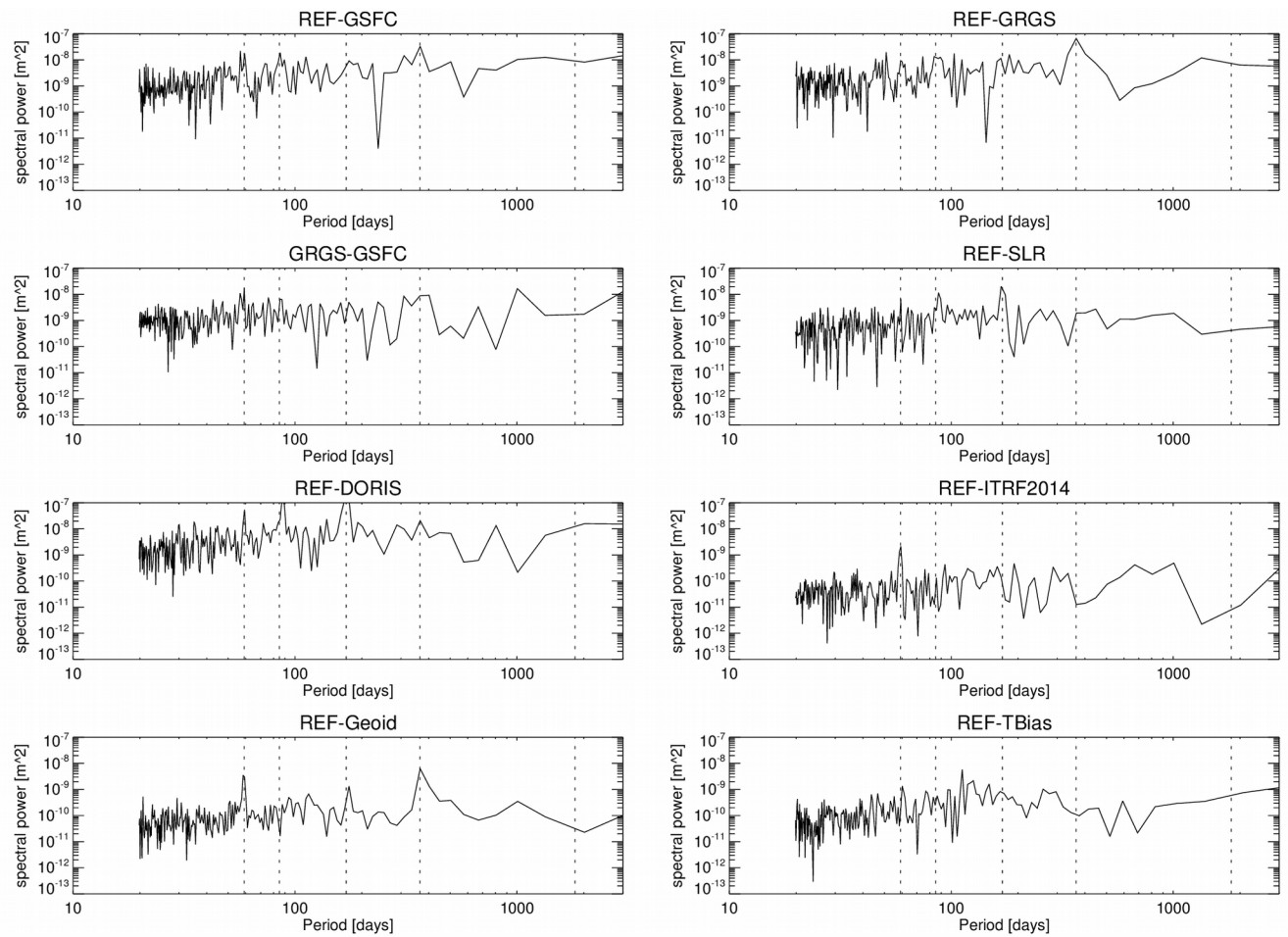

**Figure S2: Power spectra of the global mean radial orbit differences over the oceans for *REF minus*, *GSFC*, *REF minus GRGS*, GRGS *minus*, *GSFC*, *REF minus SLR*, *REF minus DORIS*, *REF minus ITRF14*, *REF minus Geoid*, and *REF minus TBias*. Vertical dashed lines mark periods of 59, 85, 170 days, 1 and 5 years.**

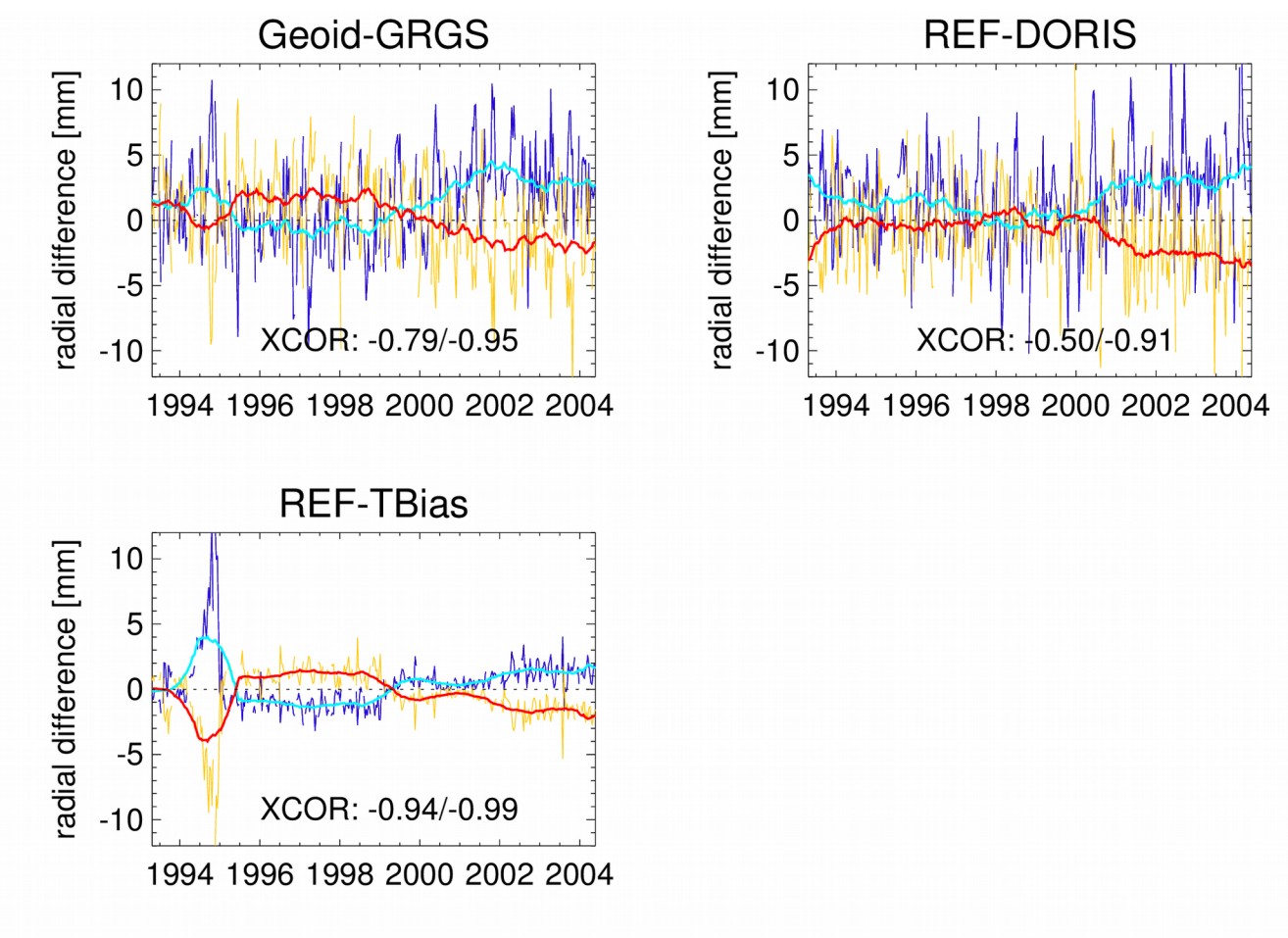

Figure S3: Global mean radial orbit differences over the oceans for *Geoid minus GRGS, REF minus DORIS*, and *REF minus Tbias* separately for ascending (blue, cyan) and descending (yellow, red) tracks and 1-year box-car filtered. The cross-correlation coefficient between the ascending and descending passes for the original and the filtered series is given at the lower part of each graph.