# Peer review of "Orbit related sea level errors for TOPEX altimetry at seasonal to decadal time scales"

_Ocean Science, 2017_

## Referee Comment (RC1) · N. P. Zelensky (Referee) · 11 Aug 2017

Referee comments Orbit related sea level errors for TOPEX altimetry at seasonal to decadal time scales os-2017-51a

I read the paper with interest. It is well written and should offer useful TOPEX altimeter satellite orbit error analysis to the altimeter/oceanographic community. I have few questions and comments, and recommend publication after these are addressed. My comments are intended to possibly help clarify and improve some aspects of an already good paper.

General Comments

\* The orbit error analysis is based on orbit differences, which exclude any error common

to the orbits. This should be explicitly mentioned in the paper as well as the authors' assumptions in using orbit differences as an error estimate.

* It seems the large and very similar REF-GRGS and REF-GSFC differences shown in Fig 5 could be better explained. Although similar in structure, the REF-Geoid plot features are so much smaller that one may exclude these gravity model differences as the primary cause of the REF-GRGS plot features. I suggest the authors include the GRGS-GSFC plot in the paper for which the orbits have much greater gravity model differences . I suspect much of the annual variability now shown is due to non-tidal station loading for the REF orbit. This contribution is not tested, but the GRGS-GSFC plot may help better identify the source of the annual variations.

* I also suggest running a spectral analysis on all the orbit differences to better identify all periodic signals; even if they are not evaluated they can be noted in the paper.

* The orbit trend differences are very small (Table 4), however Table 1 indicates substantial differences in the time variable gravity models. I suggest the authors describe/compare which gravity coefficient rates are used over the TOPEX period for all models, and which are defined by SLR.

* This my main comment. The conclusion ends with a recommendation to try to better determine low-order gravity time-varying terms past the 5x5 field employed by the GSFC orbits using improved techniques with SLR/DORIS. The recommendation is not phrased clearly since the GSFC orbits show the lowest crossover residuals compared to the other test orbits which employ 50x50 and 80x80 time varying terms (Table 1). The problem for improving time variable gravity modeling over the TOEPX period should be an issue for all the orbits tested. The authors should mention this and clarify which time-varying terms are determined with SLR for the EIGEN-6S2 and EIGEN-6S4 fields over the TOPEX period and indicate the origin of the other terms. Furthermore the GSFC and Geoid orbit slightly improved performances over the REF orbit could suggest extrapolation of the higher order GRACE-defined seasonal terms to the

TOPEX era may even be harmful. For example is the 5mm REF-Geoid annual signal (Fig 3) due to error in the REF orbit? The authors should clarify the conclusion in consideration of these remarks.

Specific Comments

* Any explanation why the DORIS residuals are slightly higher for the DORIS-only orbit? One would expect a decrease in the DORIS residuals compared to the DORIS+SLR orbit DORIS residuals.

* p7 l191 "The annual component of the global mean orbit differences is not included in Table 4, since it is not significant." Insignificant in amplitude, or is the formal error larger than the estimate?

* p7 l212-216 non-tidal station loading for the REF orbits can also be included as a "plausible source" for annual orbit difference variations.

* It is not clear what the REF-DORIS plot in Figure 5 is intended to show?

* It is interesting that the Figure 9 REF-DORIS plots show substantial trend differences for the ascending/descending passes. How is the considerable DORIS TOPEX network time bias treated for the various orbits? The SLR network should not have any significant time bias. It has been shown to be closely aligned with GPS time over the Jason-1 period (Zelensky et al. 2006, "DORIS time bias estimated using Jason-1")

* p9 l264 "The most striking feature is that the ascending trends are opposite to the descending trends." Say one orbit is always ahead of the other and will reach the North pole region (+Z) first, and the trend is positive. After reaching the North pole, the orbits will race South (-Z), and now will not the trend be negative?

---

## Referee Comment (RC2) · J. M. Huthnance (Referee) · 11 Sep 2017

Unfortunately as editor I have only one of the two reviews expected. I am not expert to review this myself, but can comment that this reads as a well-conceived study of a question that is important to confident estimation of sea-level trends and variability, regionally and globally. In turn, (trends of) sea-level statistics are an important subject scientifically and with social impact. In order to move the process on, I am writing this comment to formally complete this review stage. I do very much want to receive a second expert review and will transmit this as soon as available. Meanwhile I suggest that the authors make their response to the comments of the first reviewer and "minor revisions" suggested. If the second review does not appear "in parallel" there may have to be a short second review stage.

---

## Author Comment (AC1) · 3 Nov 2017

**Authors' response to reviewer #1**

First of all, we would like to thank the reviewer for his careful reading and the very insightful and helpful comments. In the following, the reviewers' comments are in black, followed by our replies in blue. Modified and new text is in italic. The main changes in the revised manuscript are related to more thoroughly analysis of the orbit uncertainties related to time variable gravity field modeling and an analysis of the effects of the estimation of the DORIS system time bias on the radial orbit components of ascending and descending satellite passes.

**General Comments**

* The orbit error analysis is based on orbit differences, which exclude any error common to the orbits. This should be explicitly mentioned in the paper as well as the authors' assumptions in using orbit differences as an error estimate.

We agree with this comment. In addition to the already existing phrases we have included the following statements in the sections 'Introduction', 'Methods' and 'Summary and Conclusions'.

Page 2, lines 61-62: *Note, that our assessment necessarily excludes contributions from errors common to these three orbits.*
Page 4, lines 94-95: *The approach adopted for the estimation of the radial orbit errors implicates that errors common to all three orbits can not be detected. In particular, all three orbits rely on the ITRF2008 reference frame and basically the same set of tracking stations.*
Page 7, lines 190-193: *Since the radial orbit components map directly to the derived sea level heights, we consider the differences presented here to represent estimates of the orbit related sea level error. However, since the orbit error analysis is based on orbit differences, any error common to all three orbits will be lacking in our assessment.*
Page 12, lines 354-355: *We estimate the orbit errors from the radial orbit differences which implies that errors common to all orbits can not be detected.*

* It seems the large and very similar REF-GRGS and REF-GSFC differences shown in Fig 5 could be better explained. Although similar in structure, the REF-Geoid plot features are so much smaller that one may exclude these gravity model differences as the primary cause of the REF-GRGS plot features. I suggest the authors include the GRGS-GSFC plot in the paper for which the orbits have much greater gravity model differences. I suspect much of the annual variability now shown is due to non-tidal station loading for the REF orbit. This contribution is not tested, but the GRGS-GSFC plot may help better identify the source of the annual variations.

We have included the GRGS-GSFC plot in the paper (Fig. 5) as suggested. We now interpret the results in more detail (page 9, lines 256-270) and come to the conclusion that modeling of the non-tidal station loading is a plausible source for part of the observed annual differences.

* I also suggest running a spectral analysis on all the orbit differences to better identify all periodic signals; even if they are not evaluated they can be noted in the paper.

We have included plots of the power spectra of the global mean radial orbit differences for all orbit differences in the electronical supplement (Fig. S2). We have added a short summary of the relevant periodic signals in Section 3.2.

Page 8, lines 229-233: *A spectral analysis of the global mean radial differences (Fig. S2) exhibits peaks at ~60 days for all but the GSFC and TBias orbit differences and at ~90 and ~170 days for the SLR and DORIS orbit differences. A weak annual component can be observed for the GRGS and Geoid orbit differences.*

* The orbit trend differences are very small (Table 4), however Table 1 indicates substantial differences in the time variable gravity models. I suggest the authors describe/compare which gravity coefficient rates are used over the TOPEX period for all models, and which are defined by SLR.

We have provided in Table 1 more details on the time-variable Earth's gravity field modelling, namely, for which coefficients drift (linear) terms and annual and semi-annual variations are available. Additionally, we have added a new paragraph in Section 2.1 describing the Earth's time-variable gravity modeling for the three orbits (page 3, lines 83-93). To our experience differing drift terms of time variable gravity field models mainly induce dipole like patterns of regional trend differences (Table 5, Fig. 7). On the global mean these signals tend to cancel (Table 4).

* This my main comment. The conclusion ends with a recommendation to try to better determine low-order gravity time-varying terms past the 5x5 field employed by the GSFC orbits using improved techniques with SLR/DORIS. The recommendation is not phrased clearly since the GSFC orbits show the lowest crossover residuals compared to the other test orbits which employ 50x50 and 80x80 time varying terms (Table 1). The problem for improving time variable gravity modeling over the TOEPX period should be an issue for all the orbits tested. The authors should mention this and clarify which time-varying terms are determined with SLR for the EIGEN-6S2 and EIGEN-6S4 fields over the TOPEX period and indicate the origin of the other terms. Furthermore the GSFC and Geoid orbit slightly improved performances over the REF orbit could suggest extrapolation of the higher order GRACE-defined seasonal terms to the TOPEX era may even be harmful. For example is the 5mm REF-Geoid annual signal (Fig 3) due to error in the REF orbit? The authors should clarify the conclusion in consideration of these remarks.

As mentioned in our response to the previous comment, we have specified in Table 1 and additionally described in Section 2.1 the details on modeling the Earth's time variable gravity. We further investigate and discuss the differences between the REF and the Geoid orbits at various places in more detail (page 6, lines 176-179; page7, lines 218-220; page 9, lines 269-270 and lines 285-288). We come to the conclusion that differences of the annual/semiannual terms of EIGEN-6S2 and EIGEN-6S4 for the pre-GRACE period are responsible for the enhanced performance of the Geoid with respect to the REF orbit. Differences of the regional interannual and decadal trends stem most probably from the yearly TVG terms derived from GRACE data starting from August 2002.
Additionally, we have replaced the last sentences of the Section "Summary and Conclusions" in order to clarify the conclusions (page 13, lines 409-419).

**Specific Comments**

\* Any explanation why the DORIS residuals are slightly higher for the DORIS-only orbit? One would expect a decrease in the DORIS residuals compared to the DORIS+SLR orbit DORIS residuals.

We do not think, that the DORIS residuals of the DORIS-only orbit are necessarily smaller than those of DORIS+SLR orbit. In fact, the level of SLR and DORIS residuals for DORIS+SLR orbit is refined by the weighting of observation types (SLR and DORIS). In case, if SLR observations are given higher weight than DORIS observations, the SLR residuals of the SLR-only orbit should be smaller than those of the DORIS+SLR orbit and DORIS residuals of the DORIS-only orbit should be higher than those of the DORIS+SLR orbit. This is the case one observes in Table 2. This is, in fact, what is meant in the sentence "*This is related to the weighting of observations used (3 cm for SLR and 0.05 cm/s for DORIS) and to the number of observations used.*" on page 5, lines 139-140.

\* p7 l191 "The annual component of the global mean orbit differences is not included in Table 4, since it is not significant." Insignificant in amplitude, or is the formal error larger than the estimate?

Insignificant in amplitude, we have clarified this.
Page 8, line 232: *Since the annual amplitude is less than 1 mm only, it can be neglected and is not included in Table 4.*

\* p7 l212-216 non-tidal station loading for the REF orbits can also be included as a "plausible source" for annual orbit difference variations.

We agree with this comment. We have rephrased the paragraph.
Page 9, lines 265/266: *Another plausible source of the relatively strong signal for the GSFC and GRGS orbit cases are the differences in the annual corrections for station coordinates by geocenter motion corrections and non-tidal atmospheric loading.*

\* It is not clear what the REF-DORIS plot in Figure 5 is intended to show?

We have replaced this subplot by the corresponding plot for GRGS-GSFC. The whole paragraph is rephrased and the results are discussed in more detail.

\* It is interesting that the Figure 9 REF-DORIS plots show substantial trend differences for the ascending/descending passes. How is the considerable DORIS TOPEX network time bias treated for the various orbits? The SLR network should not have any significant time bias. It has been shown to be closely aligned with GPS time over the Jason-1 period (Zelensky et al. 2006, "DORIS time bias estimated using Jason-1")

In fact, the DORIS system time bias was estimated once per arc for the REF and GSFC orbits, and no bias correction was applied for the GRGS orbit. We have added this information in Table 1 and included a description and a plot of the DORIS time bias estimated with GFZ's orbits in Section 3.2 (Fig S1). Following your suggestions we have expanded our study. We have derived and analyzed an additional test orbit without adjustment of the DORIS time system. The corresponding analyses and results are described in Section 3.4.

\* p9 l264 "The most striking feature is that the ascending trends are opposite to the descending trends." Say one orbit is always ahead of the other and will reach the North pole region (+Z) first, and the trend is positive. After reaching the North pole, the orbits will race South (-Z), and now will not the trend be negative?

We have rephrased the corresponding paragraph. Making use of the new test orbit we introduced, we observe in fact a strong relationship between the estimated DORIS time bias and the global mean radial orbit differences, anti-correlated for ascending and descending tracks. The regional pattern of the decadal trends is a coherent pattern over the Tropics and Subtropics. The same pattern shows up as leading pattern with about 60% explained variance from an EOF-analysis (see attached figure).

The DORIS system time bias is clearly related to along-track orbit position changes. A time bias of the altimeter measurement is known to induce (due to the latitude dependance of the altitude rate) radial errors which are anti-correlated for ascending and descending tracks (e.g. Scharro, R.: A Decade of ERS Orbits and Altimetry, PhD thesis, 2002, Fig. 5.1). However, the corresponding regional patterns would be different from the ones we observe, they should reveal a change of sign somewhere around the Tropics.

Even though our study shows, that uncertainties of the DORIS time bias are capable to induce ascending/descending discrepancies it does not fully explain the mechanism that transfers the along-track errors to radial orbit errors. However this question is beyond the scope of our paper and might be subject of a proper study. We have added a corresponding phrase at page 11, line 330: *This mechanism is not fully understood but a further analysis is beyond the scope of this paper.*

[Figure]

Figure: Leading pattern and time series derived from an EOF analysis of the *REF-TBias* orbit differences for ascending and descending passes.

---

## Author Comment (AC2) · 3 Nov 2017

Dear editor, we have prepared and posted the response to the comments of the first reviewer and included the "minor revisions" suggested.

———————————————

---

## Author Comment (AC3) · 3 Nov 2017

Dear editor, we have prepared and posted the response to the comments of the first reviewer and included the "minor revisions" suggested.

———————————————

---

## Referee Report (RR1)

Referee comments 2nd iteration
Orbit related sea level errors for TOPEX altimetry at seasonal to decadal time scales
os-2017-51a version3 update

The updated paper is easier to read than the original and offers additional useful information. There are however a few more questions and remarks the authors should address for improved clarity. I recommend publication after these are addressed.

**General Comments**

- My main comment is that the authors do not explain their reasoning and assumptions for labeling their orbit error estimates derived from orbit differences as "upper bound errors". Orbit differences do not include error common to the orbits and would be better suited for estimating the lower bounds to orbit error. On page 7 l200-201 the authors say "Regional upper bound errors are guessed from the corresponding maximum RMS values over the ocean at the 1°x1° grid". This explanation is far from satisfactory. What are the assumptions made such that this will represent regional upper bound orbit error? Furthermore there is no corresponding description, including assumptions, of how "global upper bound error" is estimated. The 7mm RMS "upper bound estimate" for TOPEX global radial orbit error simply shown in Table 4 seems far too optimistic. For example, the 7mm REF-GRGS RMS difference (Tab 4) can be used as an orbit error estimate if we assume that: 1) common combined error = independent combined error = 7mm, 2) the error is shared evenly between the two orbits (orbit error = $((7^{**}2+7^{**}2) / 2)^{**}(1/2)$ = 7). Even given all these assumptions a 7mm SLR+DORIS TOPEX realistic RMS radial error estimate seems much too optimistic, not to speak of a potentially much larger upper bound error estimate, since only 10mm radial accuracy (at best) has been achieved for the Jason-2/3 or other satellite orbits which carry the post-TOPEX advanced DORIS DGXX receiver (see for example Zelensky etal 2010 "DORIS/SLR POD modeling improvements for Jason-1 and Jason-2", or for example Zelensky etal 2016 "Towards the 1-cm SARAL orbit"). The authors may also consider mentioning the 1995 TOPEX orbit evaluation which estimated the radial error at 3 cm (Marshall etal 1995 "The temporal and spatial characteristics of TOPEX/POSEIDON radial orbit error")
  I suggest not to classify the error estimates as "upper bound". In any case the authors should include a paragraph, or better a small section, devoted to describing the methods and especially the assumptions for estimating "upper bound" or "lower bound" or "any another category" of global and regional orbit errors made using orbit differences. The description should include RMS, trend, and amplitude values since they are presented in the paper.

- Page 3 lines 81-83 summarize the main differences between the GFZ, GRGS, and GSFC orbits. I suggest including SLR/DORIS weighting combined with LRA modeling as an important orbit modeling difference. The SLR(cm) / DORIS(cm/sec) sigma weighting for (GFZ, GSFC, GRGS) are (30/.2, 10/.2, 6.7/.2).

Comparatively SLR data will have the most prominence in the GRGS solution, but which has the least sophisticated modeling of the LRA. Compared to the other solution data weightings, DORIS data will predominantly drive the GFZ orbit solution. I also suggest adding a row in Table 1 describing the empirical parameter estimation. For GSFC that would be : 1 Cd drag / 8-hours, 1 along-track & 1 cross-track OPR acceleration / 24-hours.

- Tables 4 - 5 are a summary of Figures 5-7 and are difficult to understand without first looking at Figures 5-7. However, the tables are presented first. If the presentation order is not changed, I suggest to at least identify the corresponding Figure in the Table labels. For example changing the Table 5 label "5-year trend (mm/year)" to "5-year trend variability (mm/year) (see Fig. 6)" would be very helpful. Such a clarification would be useful for all these tables. In addition I suggest putting: "Altimeter Crossover residuals" or "Altimeter Crossover differences" in the Table 3 column header label now empty, "Global" in the Table 4 empty label, "Regional maximum" in the Table 5 empty label. It is not clear if the global values are computed using only those regions where the formal sigma is smaller than the estimate. Are the RMS values shown in the tables the mean RMS values?

**Specific Comments**

- There must have been a mis-understanding of my question from the previous review - "Any explanation why the DORIS residuals are slightly higher for the DORIS-only orbit? One would expect a decrease in the DORIS residuals compared to the DORIS+SLR orbit DORIS residuals." . The author's response essentially said "We do not think, that the DORIS residuals of the DORIS-only orbit are necessarily smaller than those of DORIS+SLR orbit.". Yet, on page 5 l140-142 the authors write: "Among five orbits derived using DORIS observations, a slightly increased average value of DORIS RMS fits (0.04795 cm/s) is obtained for the DORIS orbit derived using only DORIS observations followed by the TBias orbit (0.04785 cm/s), while the other orbits ..."
- p 4 l118    Why are GSFC orbits listed as a correction for computing the Altimeter Crossover differences? Does not each test orbit contribute for computing the test-specific Altimeter Crossover differences?
- p7 l200    "Regional upper bound errors are guessed from" -> "Regional upper bound errors are estimated from"
- p 8 l229    "mean orbit errors"? Do you mean "mean RMS orbit errors"?
- p 8 l246-247       "The global mean decadal trends (calculated over the full mission time) are mostly significant but can be further neglected, since they are two orders of magnitude smaller than the observed sea level signal over this period (~3 mm/year)." Question – does this suggest sea level trends computed over 5-years are not reliable?
- p9 l267    "The thoroughly reflection" -> "A careful consideration"
- p9 l270    "in the pre-GRACE period." -> "in the pre-GRACE period (Fig 5)."

- p13 l399 "time-invariant annual" -> "periodic annual"
- p 21-22 Why are the REF-DORIS decadal trend signs different between Tables 4 and 6?
- It is interesting most of the decadal trend REF-Test signs are positive. The values, however are very small.
- p 29 l705 "Trend" -> "Decadal trend"

---

## Author Response (AR2)

**Authors' response to reviewer #1**

We would like the reviewer for his repeated thoroughly reading of the manuscript. In the following, the reviewers' comments are in black, followed by our replies in blue. Modified and new text is in italic.

The **main changes** besides the  wording in the revised manuscript are:
 - change of the calculation method of the RMS values given in table 4
 - correction of the time series of global mean radial orbit differences over the ocean (error in the application of the land/water mask) which implies small changes in tables 4 and 6 and figures 4, S2 and S3 and the corresponding text
 - improved description of the methods used to estimate global and regional orbit related errors

**Main Comments:**

• *My main comment is that the authors do not explain their reasoning and assumptions for labeling their orbit error estimates derived from orbit differences as "upper bound errors". Orbit differences do not include error common to the orbits and would be better suited for estimating the lower bounds to orbit error.*

Our estimates are obtained from the comparison of three independent orbit solutions. Since the three orbits were derived using various up-to-date models and reference frames, the errors common to the three orbits should be rather low which makes us confident that our error estimates represent most of the error. The following sentence has been added in the Section "Conclusion", page 13, lines 413-415: *'This error is notably less than the 3-4 cm radial orbit error obtained for TOPEX/Poseidon by Marshall et al. (1995) indicating the advance in orbit modelling for this satellite over the past 20 years.'*
The corresponding reference (Marshall et al. (1995)) has been added to the reference list.

*On page 7 l200-201 the authors say "Regional upper bound errors are guessed from the corresponding maximum RMS values over the ocean at the 1°x1° grid". This explanation is far from satisfactory. What are the assumptions made such that this will represent regional upper bound orbit error? Furthermore there is no corresponding description, including assumptions, of how "global upper bound error" is estimated.*

Please refer to the our answer given two paragraphs below.

*The 7mm RMS "upper bound estimate" for TOPEX global radial orbit error simply shown in Table 4 seems far too optimistic. For example, the 7mm REF-GRGS RMS difference (Tab 4) can be used as an orbit error estimate if we assume that: 1) common combined error = independent combined error = 7mm, 2) the error is shared evenly between the two orbits (orbit error = ((7\*\*2+7\*\*2) / 2)\*\*(1/2) = 7). Even given all these assumptions a 7mm SLR+DORIS TOPEX realistic RMS radial error estimate seems much too optimistic, not to speak of a potentially much larger upper bound error estimate, since only 10mm radial accuracy (at best) has been achieved for the Jason-2/3 or other satellite orbits which carry the post-TOPEX advanced DORIS DGXX receiver (see for example Zelensky etal 2010 "DORIS/SLR POD modeling improvements for Jason-1 and Jason-2", or for example Zelensky etal 2016 "Towards the 1-cm SARAL orbit"). The authors may also consider mentioning the 1995 TOPEX orbit evaluation which estimated the radial error at 3 cm (Marshall etal 1995 "The temporal and spatial characteristics of TOPEX/POSEIDON radial orbit error")*

The global mean RMS in table 4 had been derived from the regional RMS values at the 1°x1° grid. The interpolation of the orbit differences includes substantial spatial smoothing and merging of different passes. This corresponds to the processing of the altimeter sea itself and should make the error estimate comparable to the sea level data itself. In order to make the results more consistent and easy to

understand, we decided to change the method how to calculate the RMS value in table 4. Now, it is calculated relative to the temporal mean of the global mean difference series over the ocean in order to show the effect of radial orbit errors on mean global sea level error. Therefore, the corresponding numbers are now much smaller. They have been changed in table 4 and in the text.

*I suggest not to classify the error estimates as "upper bound". In any case the authors should include a paragraph, or better a small section, devoted to describing the methods and especially the assumptions for estimating "upper bound" or "lower bound" or "any another category" of global and regional orbit errors made using orbit differences. The description should include RMS, trend, and amplitude values since they are presented in the paper.*

Following the paper by Couhert et al. (2015) we had chosen the expression upper bound errors (their table 6). Following your comments we think this notation is arguable for the global mean error estimates and have replaced the corresponding terms by the expression 'error estimate' (p1 l15, p2 l58, p3 l68, p7 l196, p7 l199, p9, l259, p13 l398). However, for the regional errors we use the regional maximum as error estimate which justifies the use of the term 'upper bound errors'. As suggested, we have rephrased large parts of section '3.1 Methods' and state now more clearly how global and regional errors are estimated (page7, lines 212-224).

*• Page 3 lines 81-83 summarize the main differences between the GFZ, GRGS, and GSFC orbits. I suggest including SLR/DORIS weighting combined with LRA modeling as an important orbit modeling difference. The SLR(cm) / DORIS(cm/sec) sigma weighting for (GFZ, GSFC, GRGS) are (30/.2, 10/.2, 6.7/.2). Comparatively SLR data will have the most prominence in the GRGS solution, but which has the least sophisticated modeling of the LRA. Compared to the other solution data weightings, DORIS data will predominantly drive the GFZ orbit solution.*

We agree with this comment. Therefore, we added the following text on page 3, lines 87-89:
*'...as well as the constraints of the observation data (SLR/DORIS). While for the GRGS solution comparatively high weight is on the SLR data, for the GFZ solution there is higher weight on the DORIS data.'*

*I also suggest adding a row in Table 1 describing the empirical parameter estimation. For GSFC that would be : 1 Cd drag / 8-hours, 1 along-track & 1 cross-track OPR acceleration / 24-hours.*

The information on the estimation of drag coefficients and empirical accelerations has been added into Table 1, as suggested by the Reviewer.

*• Tables 4 - 5 are a summary of Figures 5-7 and are difficult to understand without first looking at Figures 5-7. However, the tables are presented first. If the presentation order is not changed, I suggest to at least identify the corresponding Figure in the Table labels. For example changing the Table 5 label "5-year trend (mm/year)" to "5-year trend variability (mm/year) (see Fig. 6)" would be very helpful. Such a clarification would be useful for all these tables. In addition I suggest putting: "Altimeter Crossover residuals" or "Altimeter Crossover differences" in the Table 3 column header label now empty, "Global" in the Table 4 empty label, "Regional maximum" in the Table 5 empty label. It is not clear if the global values are computed using only those regions where the formal sigma is smaller than the estimate.*

We have considered to change the presentation order but decided against. The tables are explained in the text in detail and some of the numbers discussed do not have corresponding figures. Instead, we have changed the header labels of the tables as suggested. The table labels have been clarified and the reader is now referred back to Section 3.1.

*Are the RMS values shown in the tables the mean RMS values?*

No, they are not. To illustrate the temporal evolution of the orbit differences we show the global mean RMS values per cycle (calculated for each time step from all radial orbit differences on the 1°x1° grid) in Fig. 3 and discuss it on p8, lines 242-250.

The RMS values given in tables 4 and 5 are calculated relative to the temporal mean of the corresponding series. For the global mean difference series (table 4) this is the temporal mean for the global mean differences series and for the regional time series it is the temporal mean of the corresponding grid point. Since we are not interested in the orbit error itself, but rather in the effect of radial orbit errors on global and regional sea level, we treat the radial orbit differences the same way as the sea level values from altimetry. We have rephrased large parts of section '3.1 Methods' and stated more clearly the differences between the two global RMS values. To stress the differences between the two global RMS values presented in section '3.2 Global mean errors', we have split the paragraph and changed the corresponding sentence at page 8, lines 239/240 to:

'*From the time series of global mean orbit differences over the oceans, RMS, annual cycle, 5-year trend variabilty, and decadal trend differences are calculated and used as estimate of the orbit related error on different time scales. These orbit errors are summarized in Table 4 for all orbit models together with the corresponding values derived from altimetric sea level anomalies.*'

**Specific Comments**

• *There must have been a mis-understanding of my question from the previous review - "Any explanation why the DORIS residuals are slightly higher for the DORIS-only orbit? One would expect a decrease in the DORIS residuals compared to the DORIS+SLR orbit DORIS residuals." . The author's response essentially said "We do not think, that the DORIS residuals of the DORIS-only orbit are necessarily smaller than those of DORIS+SLR orbit.". Yet, on page 5 l140-142 the authors write: "Among five orbits derived using DORIS observations, a slightly increased average value of DORIS RMS fits (0.04795 cm/s) is obtained for the DORIS orbit derived using only DORIS observations followed by the TBias orbit (0.04785 cm/s), while the other orbits …"*

The comment did not intend to claim there was no degradation of DORIS RMS for the DORIS orbit with respect to the REF orbit. Using SLR-only observations reduces SLR residuals, and using DORIS-only observations slightly increases DORIS residuals, as compared to the case, when both SLR and DORIS observations are used. This counter-intuitive effect is related to the weighting of SLR and DORIS observations in the GFZ solution. Therefore we have rewritten the related sentence at page 5, lines 151-155 as follows:

'*Among the five orbits derived using DORIS observations, a slightly increased average value of DORIS RMS fits (0.04795 cm/s) is obtained for the DORIS orbit derived using only DORIS observations (related to the weighting of observation types and the number of observations used) followed by the TBias orbit (0.04785 cm/s), while the other orbits derived using SLR and DORIS observations (REF, ITRF14, and Geoid) show comparable average values of DORIS RMS fits (0.04775 – 0.04778 cm/s).*'

• *p 4 l118 Why are GSFC orbits listed as a correction for computing the Altimeter Crossover differences? Does not each test orbit contribute for computing the test-specific Altimeter Crossover differences?*

The corresponding paragraph was not correct. The altimeter cross-over differences were calculated for each test orbit separately. However, for the calculation of sea level anomalies grids which we use to relate the orbit errors to the total sea level variability (Figure 9 and table 4) we selected only one set of correction values (the same set of correction models as for the crossover point statistics and the GSFC-orbits). The paragraph at page 4, lines 118-120 now reads*:*

'*These include: EOT11a ocean tides and loading tides (Savcenko and Bosch, 2012), solid earth tides following the IERS 2003 conventions, and updated GPD+ wet tropospheric corrections (Fernandes and Lazaro, 2016). The altimeter crossover differences were calculated for each test orbit separately. For the calculation of sea level anomaly grids the GSFC std1504 orbits have been selected.*'

• *p7 l200 "Regional upper bound errors are guessed from" -> "Regional upper bound errors are estimated from"*
done

• *p 8 l229 "mean orbit errors"? Do you mean "mean RMS orbit errors"?*
done

• *p 8 l246-247 "The global mean decadal trends (calculated over the full mission time) are mostly significant but can be further neglected, since they are two orders of magnitude smaller than the observed sea level signal over this period (~3 mm/year)." Question – does this suggest sea level trends computed over 5-years are not reliable?*
We estimate the orbit related error of the global mean 5-year trends to be 0.1 mm/year where sea level data itself exhibit a variability of 0.55 mm/year. The significance of the 5-year trends of the global mean sea level is questionable anyhow considering the presence of strong interannual signals related to e.g. global hydrology. However, this error should be considered for the estimation of the accelerations of global mean sea level rise.
We have added the following sentences at page 9, lines 275-277:
*'An error of this size might interfere with the estimation of global mean sea level acceleration. Hence, relative to the GFZ orbits the use of the GSFC (GRGS) orbits would result in a slightly increased (decreased) acceleration of the global mean sea level curve during the TOPEX period'*

• p*9 l267 "The thoroughly reflection" -> "A careful consideration"*
done

• *p9 l270 "in the pre-GRACE period." -> "in the pre-GRACE period (Fig 5)."*
done

• *p13 l399 "time-invariant annual" -> "periodic annual"*
done (changed accordingly at p14, line 442 and at p20, table 1).

• *p 21-22 Why are the REF-DORIS decadal trend signs different between Tables 4 and 6?*
Table 4 gives the estimated error which we derive from the absolute value of the derived decadal trends (now described in more detail in section '3.1. Methods'). The focus of table 6 is the differences between ascending and descending orbits which are multiples of the merged global values and have opposite signs. Therefore, in table 6 we provide the actual values of the trend differences. We have now stressed these differences in the table legends.

• I*t is interesting most of the decadal trend REF-Test signs are positive. The values, however are very small.*
As described in the paragraph above, in tables 4 and 5 we use the absolute values of the trend differences as error estimates of the decadal trend. In fact, the signs of decadal trend differences are predominately negative (only for *REF-ITRF14* it is positive). Since the decadal trends of the global differences are very small we do not discuss this issue any further.

• *p 29 l705 "Trend" -> "Decadal trend"*
done

[revised manuscript text omitted]